# Identification of 3′ UTR motifs required for mRNA localization to myelin sheaths in vivo

**Katie M. Yergert**[1], **Caleb A. Doll**[1], **Rebecca O'Rouke**[1], **Jacob H. Hines**[2], **Bruce Appel**[1] *

1 Department of Pediatrics, University of Colorado Anschutz Medical Campus, Aurora, Colorado, United States of America, 2 Department of Biology, Winona State University, Winona, Minnesota, United States of America

* bruce.appel@cuanschutz.edu

## Abstract

Myelin is a specialized membrane produced by oligodendrocytes that insulates and supports axons. Oligodendrocytes extend numerous cellular processes, as projections of the plasma membrane, and simultaneously wrap multiple layers of myelin membrane around target axons. Notably, myelin sheaths originating from the same oligodendrocyte are variable in size, suggesting local mechanisms regulate myelin sheath growth. Purified myelin contains ribosomes and hundreds of mRNAs, supporting a model that mRNA localization and local protein synthesis regulate sheath growth and maturation. However, the mechanisms by which mRNAs are selectively enriched in myelin sheaths are unclear. To investigate how mRNAs are targeted to myelin sheaths, we tested the hypothesis that transcripts are selected for myelin enrichment through consensus sequences in the 3′ untranslated region (3′ UTR). Using methods to visualize mRNA in living zebrafish larvae, we identified candidate 3′ UTRs that were sufficient to localize mRNA to sheaths and enriched near growth zones of nascent membrane. We bioinformatically identified motifs common in 3′ UTRs from 3 myelin-enriched transcripts and determined that these motifs are required and sufficient in a context-dependent manner for mRNA transport to myelin sheaths. Finally, we show that 1 motif is highly enriched in the myelin transcriptome, suggesting that this sequence is a global regulator of mRNA localization during developmental myelination.

## Introduction

In the central nervous system, myelin provides metabolic support and increases conduction velocity along axons. Myelin is produced by oligodendrocytes, glial cells that extend multiple long processes, and wrap layers of membrane around axons. Myelin sheaths originating from a single oligodendrocyte can vary considerably in length and thickness, suggesting that sheath growth is locally regulated [1–3]. In line with this model, the myelin transcriptome is distinct compared to the cell body [4]. For example, proteolipid protein (PLP) and myelin basic protein (MBP) are the most abundant proteins in myelin. Yet the underlying mechanisms driving their protein expression in the myelin are entirely different. *Plp* mRNA is retained in the cell body and translated at the endoplasmic reticulum and the protein is transported to myelin

**Data Availability Statement:** All relevant data are within the paper and its Supporting Information files.

**Funding:** This work was supported by US National Institutes of Health (NIH) grant R01 NS095670 and

a gift from the Gates Frontiers Fund to B.A. K.M.Y was supported by NIH (NIGMS) T32 fellowship GM008730, the Victor W. Bolie and Earleen D. Bolie Graduate Scholarship Fund, and as a RNA Scholar of the RNA Bioscience Initiative, University of Colorado School of Medicine. C.D. was supported by NIH grant (NINDS) R21 NS117886. J.H.H was supported by a National Multiple Sclerosis Postdoctoral Fellowship (FG 2024-A-1) and NIH (NIMH) fellowship T32 MN015442. The University of Colorado Anschutz Medical Campus Zebrafish Core Facility was supported by NIH grant P30 NS048154. The funders had no role in study design, data collection and analysis, decision to publish, or preparation of the manuscript.

**Competing interests:** The authors have declared that no competing interests exist.

**Abbreviations:** 3' UTR, untranslated region; AME, Analysis of Motif Enrichment; dpf, days post fertilization; F-actin, filamentous actin; FIMO, Find Individual Motif Occurrences; GO, gene ontology; MBP, myelin basic protein; MEME, Multiple Em for Motif Elicitation; PLP, proteolipid protein; RNA-seq, RNA sequencing; ROI, region of interest; RTS, RNA transport signal; smFISH, single molecule fluorescent in situ hybridization.

[5,6]. By contrast, *Mbp* mRNA is trafficked to nascent sheaths and locally translated [5–10]. This evidence supports the model that mRNAs are selectively targeted to nascent sheaths and locally translated during growth and maturation.

Transport and local translation of mRNAs are broadly utilized mechanisms for controlling subcellular gene expression. In neurons, mRNAs are subcellularly localized to axons [11–13], dendrites [14], and growth cones [15], and local translation is required for axon growth and synaptogenesis [16–19]. Frequently, mRNA localization in neurons is determined by elements within the 3' UTR [20,21]. For instance, the 3' UTR of *β-actin* contains a sequence that is recognized by the RNA binding protein ZBP1 for localization to cellular projections including growth cones, axons, and dendrites [22–26]. Neurons localize hundreds of mRNAs to different subcellular compartments, but the underlying localization elements within the transcripts are largely unknown.

Similar to neurons, oligodendrocytes localize hundreds of mRNAs to distal myelin sheaths [27], but the localization signals necessary for myelin enrichment are limited to a few mRNAs. To date, the most extensively investigated transcript in oligodendrocytes is *Mbp* mRNA. The *Mbp* 3' UTR is required for mRNA localization to myelin sheaths [28,29] and contains 2 minimal sequences including a 21-nt conserved sequence that is necessary for localization to processes in cultured mouse oligodendrocytes [30,31]. However, the minimal sequence is not required for localization in vivo, indicating that the *Mbp* 3' UTR contains clandestine localization signals [29]. The investigations into *Mbp* mRNA localization have provided significant insights into the molecular mechanisms underlying mRNA localization in oligodendrocytes. However, we know very little about the mechanisms that promote localization of the other hundreds of myelin transcripts. How are mRNAs selected for localization to myelin sheaths? Do myelin-localized transcripts share similar *cis*-regulatory elements?

Here we bioinformatically identified myelin-enriched transcripts and investigated the ability of their 3' UTR sequences to promote mRNA localization to nascent sheaths in living zebrafish. The 3' UTRs that promote myelin localization contain shared *cis*-regulatory motifs necessary for mRNA localization. Moreover, we found that the motifs are sufficient to promote localization in some, but not all, contexts. Furthermore, we identified a sequence motif that is highly enriched in the myelin transcriptome, implicating the motif as a global regulator of mRNA localization in myelinating oligodendrocytes. Together, our data support a model whereby motifs in 3' UTRs promote mRNA localization to nascent myelin sheaths.

## Results

### Quantification of mRNA within myelin sheaths of live zebrafish larvae

Although some transcripts, including *Mbp* and *Mobp* mRNA, are present in myelin [7–9,27,32], we lack information about the precise spatial distribution of myelin-enriched mRNAs in vivo. We therefore adapted the MS2 system [33] to visualize and quantify mRNA in myelinating oligodendrocytes of living zebrafish larvae. The MS2 system consists of a mRNA containing a *24xMBS* (MS2 binding sites) sequence, which forms repetitive stem loops, and MCP-EGFP (MS2 coat protein), a reporter protein that specifically binds the *24xMBS* stem loops for visualization of the mRNA via EGFP (Fig 1A). A nuclear localization signal is fused to the MCP-EGFP sequestering unbound MCP-EGFP in the nucleus thus reducing background fluorescence. To drive expression of MCP-EGFP in oligodendrocyte lineage cells, we created an expression plasmid, *sox10:NLS-tdMCP-EGFP*. Next we created *mbpa:mScarlet-Caax-24xMBS-3'UTR* to drive expression of mRNAs with various 3' UTR elements in myelinating oligodendrocytes (Fig 1A). Additionally, this plasmid also encodes expression of mScarlet-Caax, which acts as a myelin membrane reporter.

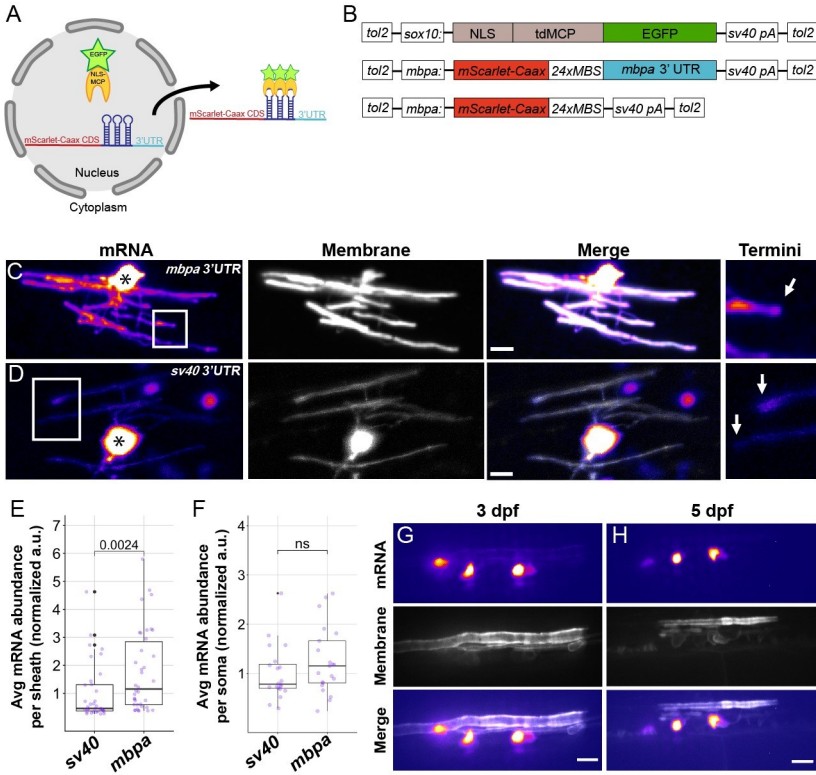

**Fig 1. The *mbpa* 3′ UTR is sufficient to localize mRNA to myelin sheaths in living zebrafish.** (A) Schematic of the MS2 system to visualize mRNA localization in oligodendrocytes. *sox10* regulatory DNA drives expression of nuclear-localized MS2 coat protein, NLS-MCP-EGFP (orange crescent and green star). *mbpa* regulatory elements drive expression of mRNA encoding mScarlet-CAAX fluorescent protein with a repetitive sequence that creates 24 stem loops (24xMBS). When co-expressed, the mRNA–protein complex is exported from the nucleus and localized via the 3′ UTR. (B) Schematic of MS2 expression plasmids used for transient expression in oligodendrocytes with target sequences for Tol2 transposase to facilitate transgene integration. (C and D) Representative images of localization directed by the *mbpa* (C) or control *sv40* 3′ UTR (D). Asterisks mark cell bodies with high expression levels of the nuclear-localized MCP-EGFP. Boxed areas are enlarged to highlight sheath termini (arrows). (E) Average mRNA abundance per myelin sheath, measured by EGFP fluorescence intensity normalized to the average intensity of the *sv40* control. *sv40*: n = 5 larvae, 35 sheaths. *mbpa*: n = 6 larvae, 38 sheaths. (F) Average mRNA abundance per soma, measured by EGFP fluorescence intensity normalized to the average intensity of the *sv40* control. *sv40*: n = 11 larvae, 20 cell bodies. *mbpa*: n = 15 larvae, 21 cell bodies. (G and H) Representative images of 2 myelinating oligodendrocytes expressing mRNA lacking the *24xMBS*. NLS-MCP-EGFP remains in the nucleus at 3 dpf (G) and 5 dpf (H). Scale bars, 10 μm. Statistical significance evaluated using Wilcoxon test. The underlying numerical data can be found in S1 and S2 Data. 3′ UTR, 3′ untranslated region; dpf, days post fertilization.

As proof of principle, we first tested the 3′ UTR of *mbpa*, a zebrafish ortholog of *Mbp*, which promotes mRNA localization in myelin (29). As a control, we used the *sv40* polyadenylation signal, which lacks any known localization signals [34,35] (Fig 1B). To examine mRNA localization in individual oligodendrocytes, we transiently expressed *sox10:NLS-tdMCP-EGFP* with either *mbpa:mScarlet-Caax-24xMBS-mbpa 3′UTR* or *mbpa:mScarlet-Caax-24xMBS-sv40 3′UTR* by microinjection into 1-cell stage zebrafish embryos. This approach revealed mRNA, via EGFP fluorescence intensity, in the cytoplasm of nascent sheaths at 4 days post fertilization (dpf). Consistent with previous reports, we found that the *mbpa* 3′ UTR was sufficient to localize mRNA to nascent sheaths in vivo (Fig 1C and 1D). Furthermore, this approach demonstrated active translation of the *mScarlet-Caax* reporter mRNA. We also observed differences in the mScarlet-Caax fluorescence intensity, at the protein level, between the *sv40* and *mbpa* 3′

UTR constructs, which could be explained by 3′ UTR-mediated difference in translation efficiency (Fig 1C and 1D).

To quantify mRNA abundance, we measured the average fluorescence intensity of EGFP in myelin sheaths. Due to high levels of fluorescent signal emitting from the cell body, we measured small regions (7 μm) of myelin sheaths far from the cell body. We found that the average fluorescence intensity of sheaths expressing the *mbpa* 3′ UTR were approximately 2-fold greater than the control (Fig 1E). Importantly, the difference in mRNA localization to myelin sheaths was not due to variability in expression levels of the MS2 reporter (Fig 1F). To verify that cytoplasmic fluorescence is mediated through the mRNA, we removed the *24xMBS* and found that EGFP was retained in the nucleus throughout developmental myelination (Fig 1G and 1H). Together, these experiments confirm the ability to visualize and quantify mRNA localization during developmental myelination in vivo.

### *mbpa* mRNA localizes to the leading edge of developing myelin sheaths

Previously, *mbpa* transcripts have been detected in sheaths throughout developmental myelination [29,36,37], but the proportion of mRNA that is transported to myelin sheaths is unknown. To quantify the distribution of endogenous *mbpa* mRNA localization in cell bodies and myelin sheaths, we performed single molecule fluorescent in situ hybridization (smFISH) on *Tg(mbpa:egfp-caax)* larvae, which express membrane-tethered EGFP-CAAX in the myelin tracts of the larval hindbrain (Fig 2A). As a control, we also detected *egfp* mRNA encoded by the transgene, which does not contain any known mRNA localization signals (Fig 2B). To quantify mRNA abundance, we calculated the average integrated density of each transcript in both oligodendrocyte cell bodies and in comparable volumes of myelin in the hindbrain. *mbpa* mRNA abundance in myelin sheaths significantly increased between 3 and 4 dpf before reaching a plateau at 5 dpf (Fig 2C), indicating that the majority of *mbpa* mRNA is transported to myelin sheaths at 4 dpf. Specifically, we found that 37% of *mbpa* transcripts localized to myelin sheaths at 4 dpf, whereas only 4% of the *egfp* transcripts localized to the myelin (Fig 2D). We therefore performed all subsequent experiments at 4 dpf, during the peak of active *mbpa* mRNA transport. At 4 dpf, the density of *mbpa* mRNA and sheath length were positively correlated indicating that longer sheaths have more *mbpa* mRNA (Fig 2E). We interpret these data to mean that longer nascent sheaths have more cytoplasmic space and therefore have a greater capacity to accumulate macromolecules such as mRNA. In support of this interpretation, previous reports have found that longer myelin sheaths have increased mitochondrial densities [38].

The improved spatial resolution of our smFISH and MS2 approaches allowed us to examine subsheath localization of mRNA within nascent sheaths. We therefore examined the distribution of single *mbpa* transcripts from both longitudinal (Fig 3A and 3B) and transverse (Fig 3C) orientations using smFISH. This revealed transcripts as discrete puncta distributed along the length of individual sheaths (Fig 3A) and at sheath termini (Fig 3B), consistent with live-imaging observations using the MS2 system (Fig 1C). This distribution was reminiscent of filamentous actin (F-actin) at the leading edge of myelin sheaths [39–41]. To determine if mRNA is localized at the leading edge of myelin membrane during wrapping, we co-expressed the MS2 system and Lifeact-mNeonGreen, an F-actin reporter. We found that transcripts containing the *mbpa* 3′ UTR colocalized with F-actin (Fig 3D), indicating that mRNA occupies the leading edge of myelin sheaths.

To determine the frequency at which mRNA localizes to the leading edge, we used the MS2 system to quantify the number of sheath termini that are enriched with mRNA. We found that 47% of sheath termini have *mbpa* 3′ UTR-containing mRNA in comparison to 27% of the *sv40*

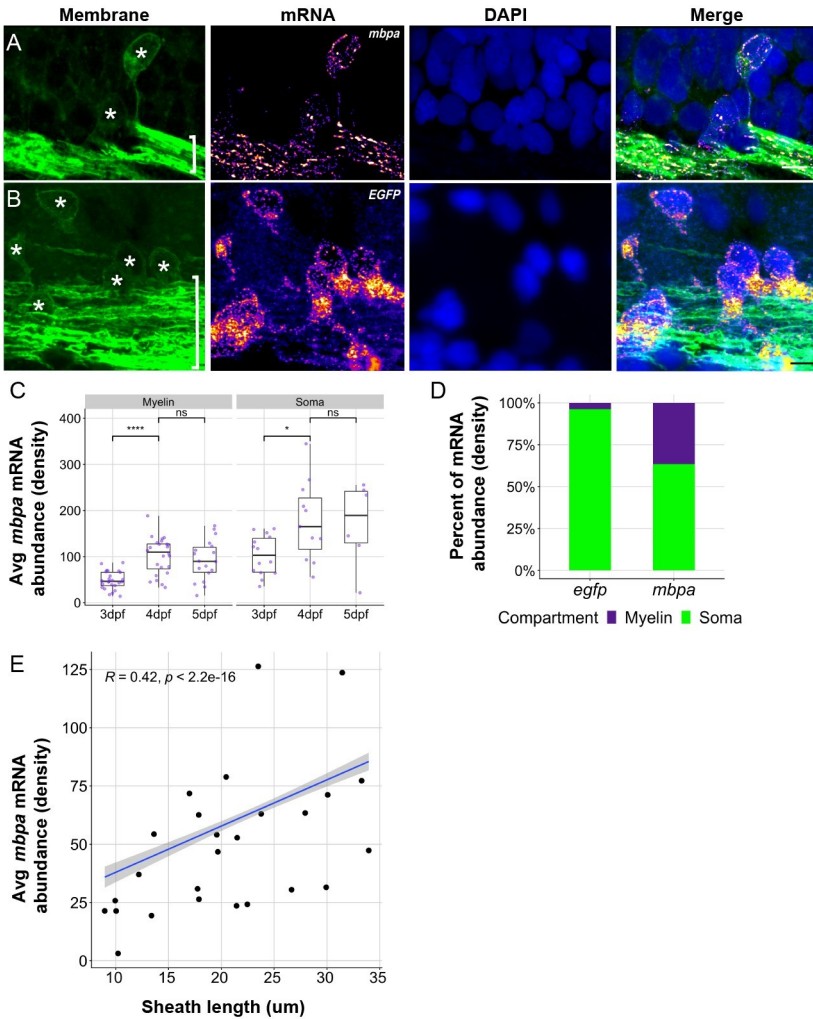

**Fig 2. Endogenous *mbpa* mRNA localizes to myelin sheaths between 3–5 dpf.** (A and B) Representative images of smFISH experiments using 4 dpf transgenic larva expressing EGFP-CAAX to mark oligodendrocytes. Images show sagittal sections of the hindbrain. DAPI stain labels nuclei. Sections were treated with smFISH probes designed to detect *mbpa* (A) or *egfp* (B) mRNA. Asterisks mark cell bodies and brackets mark myelin tracts. Scale bars, 10 μm. (C) Average *mbpa* mRNA density per cell body or equivalent volume of myelin from 3 to 5 dpf. Density was measured using the integrated density of fluorescence intensity in cell bodies and approximately equal volumes of myelin along the myelin tracts. A minimum (n) for each group was 3 larvae, 6 cell bodies, and 15 myelin regions. Statistical significance evaluated using Wilcoxon test. (D) Proportion of *egfp* or *mbpa* mRNA abundance in cell bodies compared to myelin tracts. A minimum (n) for each group was 3 larvae, 11 cell bodies, and 21 myelin regions. (E) Average *mbpa* mRNA density within individual sheaths plotted as a function of sheath length. Statistical significance evaluated using Spearman's correlation coefficient. Shaded area represents 95% confidence interval. *n* = 7 embryos, 26 sheaths. The underlying numerical data can be found in S5–S7 Data. dpf, days post fertilization; smFISH, single molecule fluorescent in situ hybridization.

3′ UTR control (Fig 3E). To precisely define the spatial organization of mRNA at sheath termini, we measured the fluorescence intensity of the MS2 mRNA reporter system across a 7-μm distance at the ends of each sheath. We found that mRNA containing the *mbpa* 3′ UTR was significantly enriched within 2 μm of the terminal end (Fig 3F). However, mRNA containing the *sv40* 3′ UTR was uniformly distributed along the length of the sheath and lacked enrichment at the leading edge (Figs 1D and 3F). Our data support the conclusion that the *mbpa* 3′ UTR is sufficient to localize mRNA to the leading edge of nascent sheaths during developmental myelination.

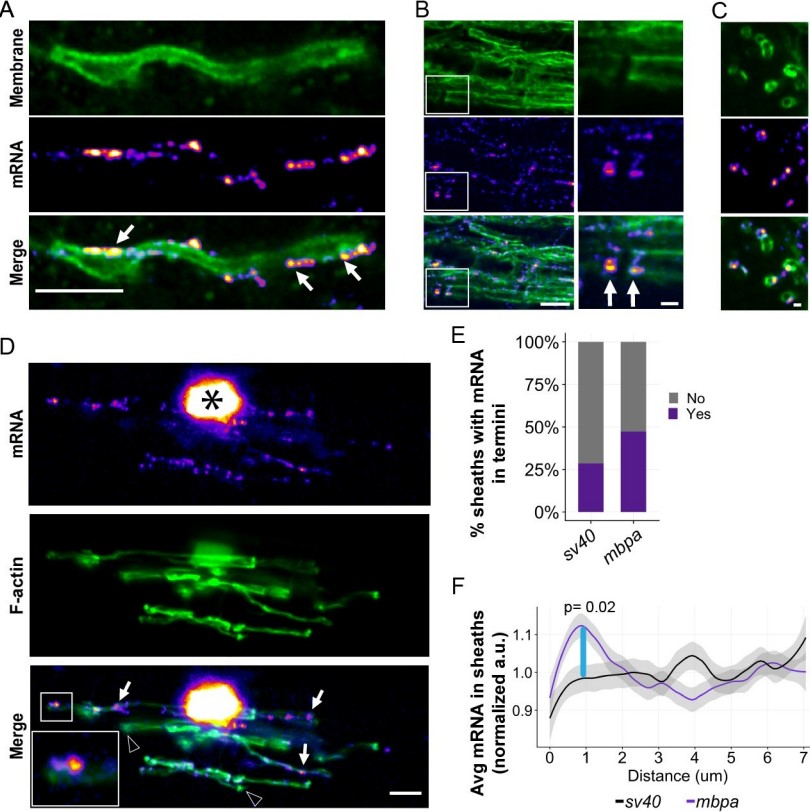

**Fig 3. The *mbpa* 3′ UTR is sufficient to localize mRNA to the leading edge of myelin sheaths during wrapping.** (A) smFISH images of a single optical section of a myelin sheath in a 3-dpf larva spinal cord. *mbpa* transcripts line the myelin sheath. Arrows highlight clusters of *mbpa* mRNA transcripts. (B) smFISH images of a single optical section of myelin tracts in the hindbrain of a 5-dpf larva. Boxed area magnified to highlight sheath termini (arrows). (C) smFISH images of a single optical section in transverse plane of myelin sheaths in a 5-dpf larva midbrain. Scale bars (A, B, and D), 5 μm; (C, boxed enlargements), 1 μm. (D) Representative images from MS2 system showing colocalization of mRNA containing *mbpa* 3′ UTR and F-actin in a myelinating oligodendrocyte. Asterisk marks the cell body, and boxes are magnified to highlight sheath termini. Arrows highlight sheaths with mRNA, and arrowheads highlight sheaths lacking mRNA. (E) Proportion of sheaths with mRNA enriched in sheath termini at 4 dpf using the MS2 system. Proportion measured as (sheaths with enrichment / number of sheaths) = 10/35 *sv40*, 18/38 *mbpa*. (F) Average fluorescence intensity of MS2 mRNA reporter containing the *sv40* or *mbpa* 3′ UTRs across a 7-μm distance, at 0.2-μm intervals, from myelin sheath termini at 4 dpf. Each line scan was normalized to the average fluorescent intensity per sheath. All normalized values for each distance were then averaged. Shaded area represents 95% confidence interval. Statistical significance was evaluated every 0.2 μm using Wilcoxon test, and the distance between 0.8–1.0 μm was statistically significant (blue line). *sv40* 3′ UTR *n* = 5 larvae, 35 sheaths. *mbpa* 3′ UTR *n* = 6 larvae, 38 sheaths. The underlying numerical data can be found in S1 and S2 Data. 3′ UTR, 3′ untranslated region; dpf, days post fertilization; F-actin, filamentous actin; smFISH, single molecule fluorescent in situ hybridization.

## mRNA localization to myelin sheaths is determined by unique 3′ UTR motifs

Our data corroborate previous work demonstrating the sufficiency of the *Mbp* 3′ UTR in mRNA localization to myelin. Do other myelin-localized transcripts utilize 3′ UTR-dependent mechanisms for localization? To investigate this question, we bioinformatically identified 6 candidate 3′ UTRs from myelin-localized transcripts (Fig 4A). Specifically, we selected candidate 3′ UTRs by filtering RNA sequencing data obtained from purified myelin isolated from P18 mouse brain [27] for the gene ontology (GO) terms oligodendrocyte, myelin, translation, and synapse. We used the latter 2 terms because we are interested in the possibility that features of myelin plasticity are similar to synaptic plasticity [42]. We narrowed the candidate

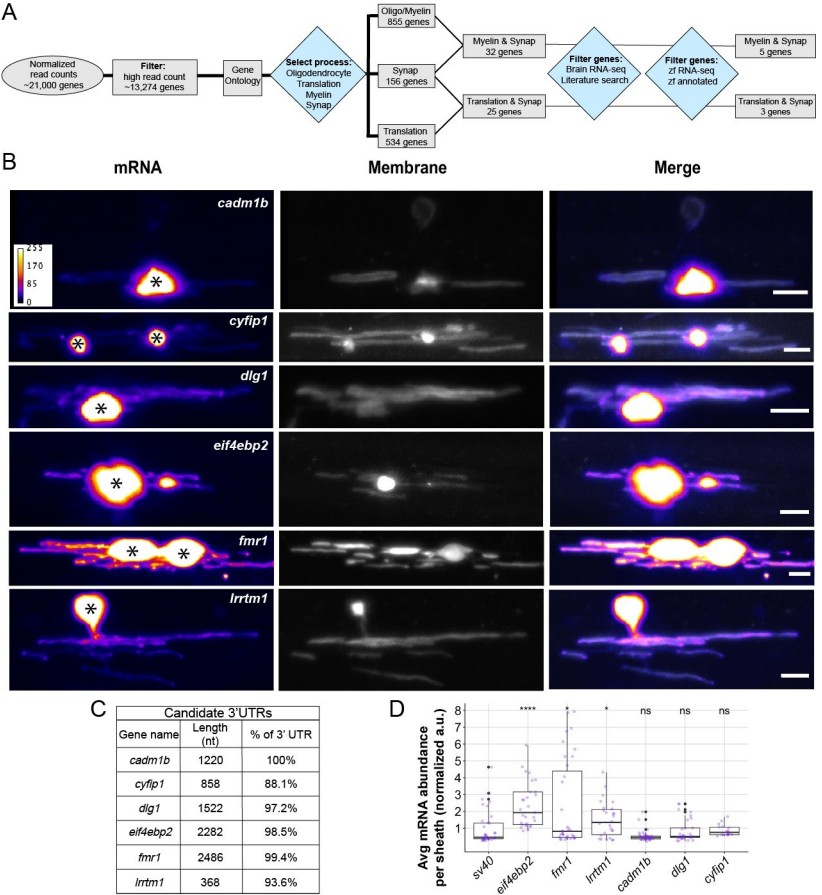

Fig 4. Different 3′ UTRs have distinct effects on mRNA localization to myelin sheaths. (A) Work flow to identify 3′ UTR candidates from RNA-seq data [27,43,44]. (B) Representative images from MS2 system showing localization of mRNAs containing different 3′ UTR sequences in oligodendrocytes. Asterisks mark cell bodies. Scale bars, 10 μm. (C) Table listing candidate 3′ UTRs incorporated into the MS2 system, 3′ UTR length, and the percentage of sequence that was cloned based on the annotated genome (GRCz11). (D) Average mRNA abundance, measured by average EGFP fluorescent intensity, per myelin sheath for each 3′ UTR. Normalized to *sv40* control, statistical significance evaluated using Wilcoxon test. A minimum (n) of 5 larvae and 18 sheaths were used in each condition at 4 dpf. The underlying numerical data can be found in S1 and S2 Data. 3′ UTR, 3′ untranslated region; dpf, days post fertilization; RNA-seq, RNA sequencing.

genes by expression levels in oligodendrocyte lineage cells from published RNA sequencing (RNA-seq) datasets [43,44], descriptions of gene functions from literature searches, and identification of zebrafish orthologs. This pipeline identified 6 candidate genes for which we cloned the 3′ UTR sequences: *cadm1b*, *cyfip1*, *dlg1*, *eif4ebp2*, *fmr1*, and *lrrtm1* (Fig 4C).

Using the MS2 system, we found that inclusion of 3′ UTRs from our candidate genes led to a wide variation in mRNA localization to nascent sheaths. Strikingly, the 3′ UTRs from *eif4ebp2*, *fmr1*, and *lrrtm1* produced significantly greater levels of fluorescence intensities in myelin sheaths than the *sv40* control, whereas the remainder, *cadm1b*, *cyfip1*, and *dlg1* were similar to the *sv40* control (Fig 4B–4D). Given that all 6 candidate transcripts are found in purified myelin, our data suggest that only a subset of myelin transcripts are localized by their 3′ UTRs and that other transcripts likely utilize *cis*-regulatory elements not present in the 3′ UTR or, alternatively, are passively localized to myelin. Nonetheless, these data expand the repertoire of 3′ UTR-dependent mRNA localization to myelin sheaths in vivo.

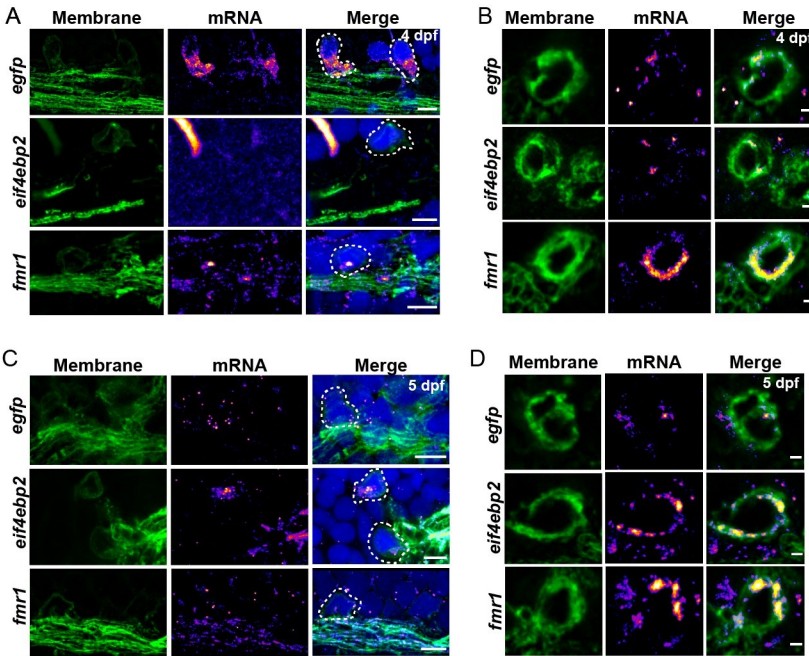

**Fig 5. *eif4ebp2* and *fmr1* mRNA are localized to sheaths during developmental myelination.** Representative images of smFISH experiments to visualize *egfp*, *eif4ebp2*, or *fmr1* mRNA localization at 4 dpf (A and B) and 5 dpf (C and D) in sagittal sections of hindbrain (A, C) or transverse sections of the Mauthner axon in the spinal cord (B, D). Dashed lines outline cell bodies marked by EGFP-CAAX. Scale bars, 5 μm (A, C) or 1 μm (B, D). dpf, days post fertilization; smFISH, single molecule fluorescent in situ hybridization.

To validate the MS2 findings, we confirmed that endogenous transcripts of *eif4ebp2* and *fmr1* are expressed by oligodendrocytes and are localized to myelin. We chose these transcripts because the 3′ UTRs are highly enriched in nascent sheaths (Fig 4D), and they encode translational regulators that are necessary for proper myelination [45] and cognition [46–48]. To investigate the spatiotemporal expression of endogenous *fmr1* and *eif4ebp2* transcripts, we used smFISH on *Tg(mbpa:egfp-caax)* larvae to label oligodendrocyte cell bodies and myelin tracts during developmental myelination. In line with the MS2 data, we observed endogenous *fmr1* expression in the cell bodies and myelin sheaths between 4 and 5 dpf (Fig 5A–5D). In contrast, *eif4ebp2* had minimal expression in oligodendrocytes at 4 dpf (Fig 5A) but was prominent in both cell bodies and myelin sheaths by 5 dpf (Fig 5C and 5D). Together, our data show that *fmr1* and *eif4ebp2* transcripts are localized to myelin sheaths, at least in part, by their 3′ UTRs.

## Localized 3′ UTRs share sequence motifs that are required for mRNA localization

3′ UTRs frequently contain regulatory elements necessary for posttranscriptional regulation including subcellular localization [20,21]. Therefore, we hypothesized that the candidate 3′ UTRs share *cis*-regulatory elements that promote localization to myelin. To test this hypothesis, we used Multiple Em for Motif Elicitation (MEME) suite bioinformatics software [49,50] to identify shared motifs among the *mbpa*, *eif4ebp2*, and *fmr1* 3′ UTRs from the annotated zebrafish genome. We identified 3 motifs shared between the candidate 3′ UTRs (Fig 6A), which correspond to unique primary sequences in each 3′ UTR. However, the *mbpa* 3′ UTR we isolated from zebrafish cDNA is truncated by 118 nucleotides at the 3′ end and does not contain

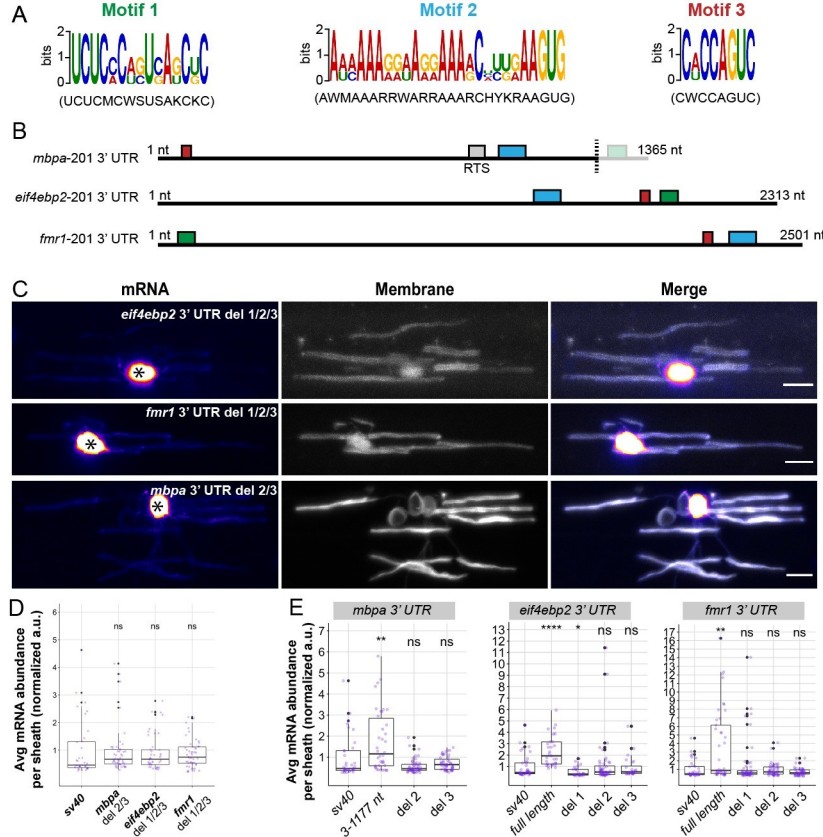

**Fig 6. Common motifs in candidate 3′ UTRs are required for myelin localization.** (A) Schematic representation of the 3 motifs identified in the 3′ UTRs of *mbpa-201*, *eif4ebp2-201*, and *fmr1-201* from the annotated zebrafish genome (GRCz11) using MEME suite bioinformatics software. (B) Schematic representation showing the relative position of the motifs within the 3′ UTRs. Green box is motif 1, blue box is motif 2, and red box is motif 3. Gray box is the conserved RTS previously identified as a minimal localization element necessary for *Mbp* mRNA transport in cultured oligodendrocytes [28,29,51]. Motif 1 in the *mbpa* 3′ UTR is not present in 3′ UTR isolated from zebrafish cDNA utilized in experimental procedures (3′ end of the dashed line). (C) Representative images of MS2 system after sequential deletions of all motifs from *mbpa*, *eif4ebp2*, and *fmr1* 3′ UTRs. Asterisks mark cell bodies. (D) Quantification of mRNA abundance in myelin sheaths from sequential deletions in (C). (E) Quantification of mRNA abundance in myelin sheaths from full length 3′ UTR, *mbpa* 3′ UTR variant 3–1177, or individual motif deletions. Statistical analysis evaluated with Wilcoxon test. Scale bars, 10 μm. A minimum (n) of 6 embryos and 35 sheaths were used in each condition (D and E). The underlying numerical data can be found in S1 and S4 Data. 3′ UTR, 3′ untranslated region; MEME, Multiple Em for Motif Elicitation; RTS, RNA transport signal.

motif 1. Additionally, the *mbpa* 3′ UTR contains a previously identified, conserved RNA transport signal (RTS) [29,51]. However, motifs 2 and 3 do not overlap with the RTS element, suggesting these motifs are novel localization elements (Fig 6B). Importantly, the identified motifs were absent from the nonenriched 3′ UTRs (*cadm1b*, *cyfip1*, and *dlg1*) and the *lrrtm1* 3′ UTR. To test requirements for these motifs, we deleted all the sequences corresponding to the identified motifs from each 3′ UTR and examined mRNA localization using the MS2 system. We found that deletion of all motifs restricted mRNA to the cell bodies (Fig 6C and 6D). To narrow down which of the motifs are required for localization, we individually deleted each sequence and found that removal of any sequence reduced mRNA localization in myelin (Fig 6E). Moreover, deletion of motif 1 from the *eif4ebp2* 3′ UTR reduced mRNA localization further than the control. Together, these data show that each motif is required for mRNA localization to nascent sheaths.

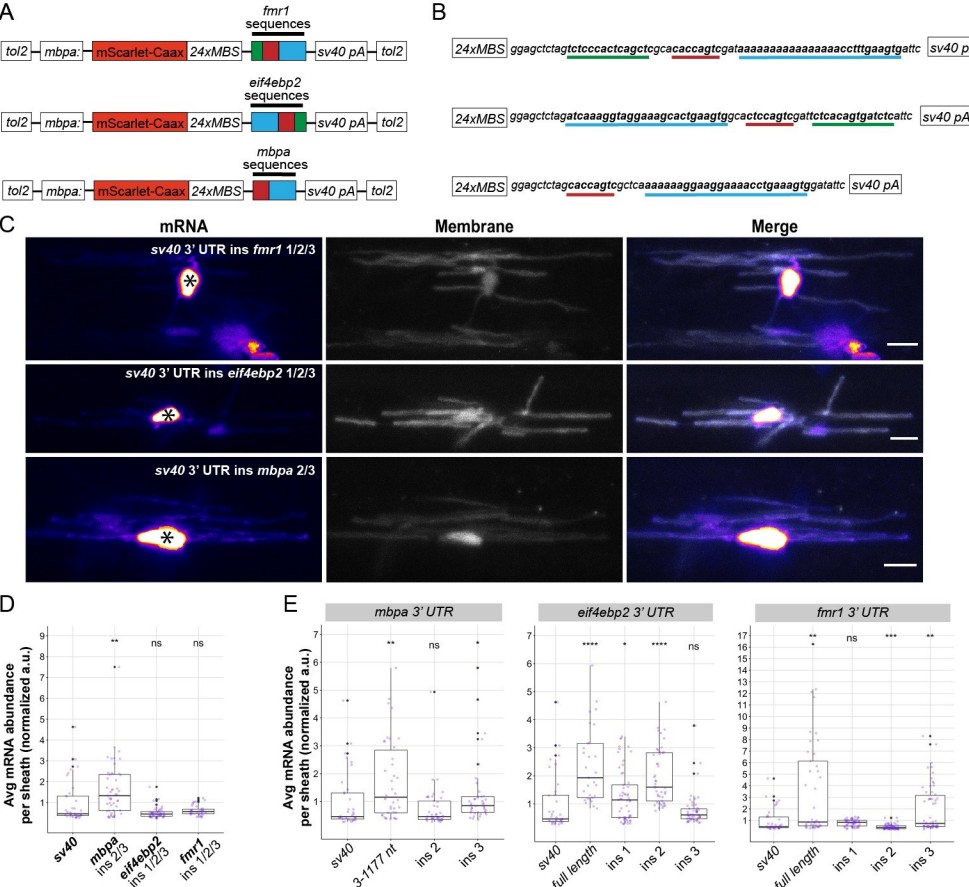

**Fig 7. Sequence motifs derived from the *mbpa* 3′ UTR are sufficient for mRNA localization to myelin.** (A) Schematic representation of the MS2 mRNA reporter with motifs inserted upstream of the *sv40* 3′ UTR. Green box is motif 1, blue box is motif 2, and red box is motif 3. (B) Schematic representation of the primary sequences used to test the sufficiency of the motifs in (A). Green underline is motif 1, blue underline is motif 2, and red underline is motif 3. Top to bottom correspond to *fmr1*, *eif4ebp2*, and *mbpa* sequences. (C) Representative images of MS2 system after motifs from the *mbpa*, *eif4ebp2*, or *fmr1* 3′ UTRs were inserted into the MS2 mRNA reporter. Asterisks mark cell bodies. (D) Quantification of mRNA abundance in myelin sheaths from (A–C). (E) Quantification of mRNA abundance in myelin sheaths from individual insertions of each motif. Statistical analysis evaluated with Wilcoxon test. Scale bars, 10 μm. A minimum (n) of 6 embryos and 35 sheaths were used in each condition (D and E). The underlying numerical data can be found in S1 and S4 Data. 3′ UTR, 3′ untranslated region.

We next tested whether the shared motifs are sufficient to localize mRNA to myelin sheaths. We cloned the unique sequences from each candidate 3′ UTR upstream of the sv40 polyadenylation signal (Fig 7A and 7B) and examined mRNA expression in oligodendrocytes using the MS2 system. We found that insertion of *mbpa*-derived sequence motifs were sufficient to localize mRNA to myelin sheaths (Fig 7C and 7D). However, insertion of the sequences isolated from *fmr1* or *eif4ebp2* were not sufficient to localize mRNA to myelin (Fig 7C and 7D).

To test the sufficiency of individual sequences, we inserted each sequence flanked by 6 to 10 nucleotides on each side upstream of the *sv40* 3′ UTR. We found that 4 out of the 8 sequences, including *mbpa* motif 3, *fmr1* motif 3, *eif4ebp2* motif 1, and *eif4ebp2* motif 2, were sufficient to localize the reporter mRNA to nascent sheaths. Together the variability in our data indicates the motifs are sensitive to contextual features intrinsic to the *cis*-elements. For instance, primary sequences [52] and secondary structures [53] frequently modulate RNA regulatory mechanisms [54,55]. Here, we find that the sufficiency of each motif to drive mRNA

localization to myelin is context-dependent. For instance, *eif4ebp2* motifs 1 and 2 are sufficient for mRNA localization independently (Fig 7E); however, in the context of motif 3, mRNA localization is repressed (Fig 7D). Furthermore, when *mbpa* motifs 2 and 3 are coupled, they are sufficient for localization (Fig 7D). However, when tested independently, we found that motif 3 accounts for the majority of the mRNA localization (Fig 7E), indicating that motif 2 does not repress mRNA localization (Fig 7D). Overall, these data highlight the complexities of mRNA localization and add to the growing body of evidence that interactions between *cis*-regulatory elements and *trans*-acting factor are context-dependent [52–57].

## Localization motifs are enriched in the myelin transcriptome

Our analyses revealed motifs shared between localized transcripts. Are these motifs commonly found in the myelin transcriptome? We cross-referenced myelin-localized transcripts with those from oligodendrocytes using 2 independent RNA-seq datasets to exclude nonoligodendrocyte transcripts [44,58]. We identified 1,855 transcripts localized to myelin (S1 Table) of which 1,771 had significantly higher expression in the myelin transcriptome and were fully annotated in the genome browser. We found that motifs 1 and 3 were not enriched in these transcripts compared to randomized, length-matched control sequences. However, motif 2 was significantly enriched in the 1,771 transcripts of the myelin transcriptome (Fig 8A). Specifically, 42.4% (751 mRNAs) of myelin-localized transcripts contain one or more copies of motif 2 (Fig 8B, S2 Table). By comparison, we found that motif 2 is present at least once in only 28.7% of the mouse transcriptome (Fig 8C, S3 Table). Localization motifs are frequently positioned in the 3′ UTRs of mRNA. We found that 63.8% of the motif 2 sites are positioned in the 3′ UTRs of myelin-localized transcripts (Fig 8D, S4–S6 Tables). Together, these data suggest that motif 2 is a localization signal utilized by many transcripts for 3′ UTR-mediated localization to myelin.

To determine if motif 2 is represented in particular subset of mRNAs within the myelin transcriptome, we performed GO analysis on all myelin-localized transcripts as well as the subset of myelin-localized transcripts containing motif 2. Previous reports investigating the myelin transcriptome show enrichment of biological processes such as nervous system development, cellular respiration, and neurogenesis (27). Assessment of our 1,771 myelin-localized genes was consistent with previous reports in that we also identified biological processes and cellular component terms pertaining to mitochondria, electron transport chain, and oxidative reduction. Importantly, we also identified GO terms associated with myelination such as myelin sheath, membrane, and protein transport (Fig 8E, S7 Table). Next, we narrowed our gene list to the subset of myelin-localized transcripts that contain motif 2 (751 genes) to determine if these genes are functionally distinct. We identified biological mechanisms associated with synaptic signaling, nervous system development, and regulation of cellular projections. Interestingly, many of the genes are associated with cellular functions in distal projections such as postsynaptic density, synaptic vesicles, axon, dendrite, and terminal bouton (Fig 8F, S8 Table). Many of the biological functions are neuronal in nature with a remarkable lack of terms associated with myelination. These observations raise the possibility that nascent myelin sheaths engage in molecular mechanisms during axon wrapping that are similar to synaptogenic mechanisms. Overall, these findings implicate motif 2 as a regulatory element for a distinct cohort of transcripts within myelin sheaths.

## Discussion

The molecular mechanisms underlying myelin sheath growth are not well understood. Purified myelin contains hundreds of mRNAs [27], lending the possibility that mRNA localization and local translation promote sheath growth and maturation. How are mRNAs selectively

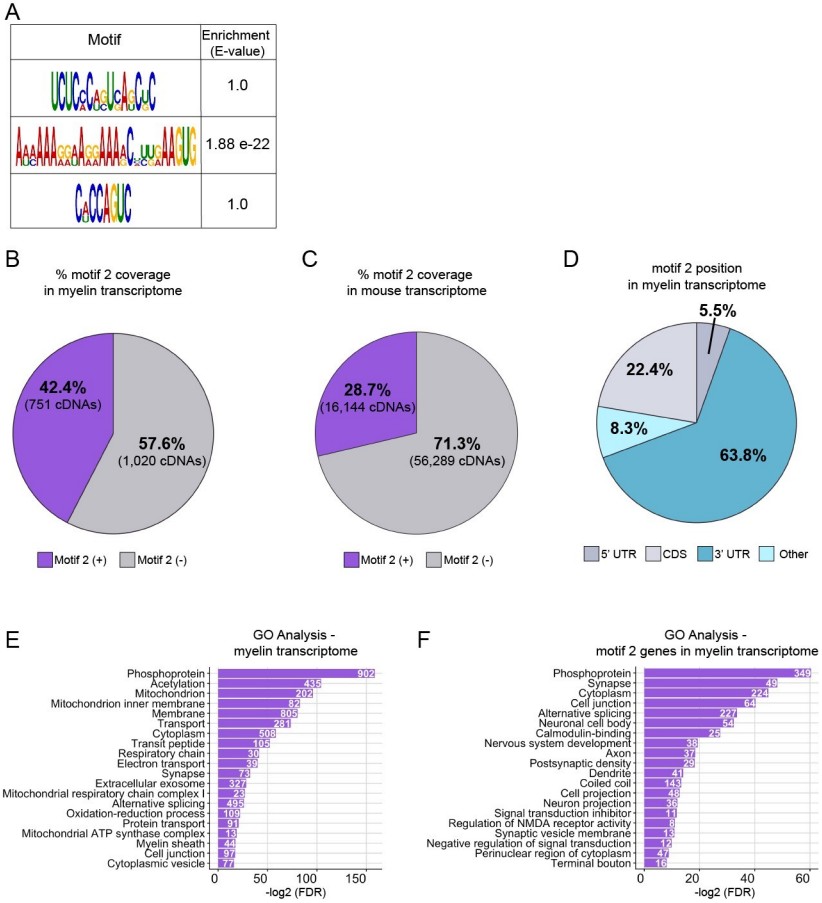

**Fig 8. Motif 2 is enriched in the mouse myelin transcriptome.** (A) Schematic representation of motifs 1, 2, or 3 enrichment in the myelin transcriptome in comparison to length-matched, randomized sequences using MEME suite Analysis of Motif Enrichment (version 5.1.1) [59]. (B) FIMO (version 5.1.1) was used to determine the frequency at which motif 2 is present in the myelin transcriptome [60]. cDNA sequences from the myelin transcriptome were analyzed for the presence of motif 2. One or more copies of motif 2 were present in 42.4% of myelin cDNAs. (C) FIMO was used to determine the frequency at which motif 2 is present in the mouse transcriptome. cDNA sequences from the mouse annotated genome (mm10) were analyzed for the presence of motif 2. One or more copies of motif 2 were present in 28.7% of mouse cDNAs. (D) Percentage of motif 2 occurrences in 5′ UTR, coding sequences, 3′ UTR or other positions in myelin transcriptome. Motifs in the "other" category represent motifs overlapping 2 regions. (E) Top 20 GO terms identified in the myelin transcriptome. (F) Top 20 GO terms identified in myelin transcripts containing motif 2. Terms are ordered from most to least significant based on -log2 of the false discovery rates. Counts represent number of genes identified with the GO term. The underlying numerical data can be found in S8–S10 Data. FIMO, Find Individual Motif Occurrences; GO, gene ontology; MEME, Multiple Em for Motif Elicitation; UTR, untranslated region.

targeted to myelin sheaths? Here we show that *cis*-regulatory elements found in candidate 3′ UTRs are required for mRNA localization to myelin during ensheathment of target axons in vivo. In particular, 1 sequence is enriched in the myelin transcriptome implicating the motif as a potential regulator of mRNA localization in oligodendrocytes.

With high-resolution microscopy, we found *mbpa* mRNA concentrated near the growth zones of nascent myelin membrane. What accounts for enrichment near growth zones? For the last several decades, the mechanisms underlying *MBP* mRNA localization have been heavily investigated and revealed that transcripts are actively transported through oligodendrocyte processes to myelin sheaths in kinesin and dynein-dependent manners [36,61,62]. Consistent with these data, recent work showed that microtubules are present in nascent

sheaths in vivo [63]. Nascent membrane also contains F-actin, which is specifically present in the leading edge [40,41]. Importantly, actin-based transport can localize mRNA, mediated by myosin motor proteins [33,64,65], but this has yet to be tested in myelinating oligodendrocytes in vivo. Here, we show that mRNA containing the *mbpa* 3′ UTR is colocalized with F-actin in myelin sheaths. These observations raise the possibility that *Mbp* mRNA is handed off from microtubule-based transport to actin-based transport within growing sheaths.

Hundreds of mRNAs are localized to myelin, but it is unclear if these transcripts utilize 3′ UTR-dependent localization mechanisms. Of the candidates we tested, 4 out of seven 3′ UTRs were sufficient to drive mRNA into nascent sheaths. Thus, for some mRNAs, *cis*-regulatory elements important for mRNA localization might be embedded in 5′ UTRs, coding regions, or retained introns of the transcripts [66–68]. Alternatively, transcripts might occupy myelin by diffusion. Notably, we identified 2 transcripts that utilize their 3′ UTRs for localization to myelin, *eif4ebp2*, and *fmr1*, which encode translational regulators. Importantly, *fmr1* mRNA and the encoded protein, FMRP, have precedence for being localized to subcellular compartments, such as dendritic spines and myelin sheaths, far from the cell body [45,69]. By finding mRNAs encoding translation regulators in myelin, our data support the possibility that transcripts encoding translational proteins are themselves locally translated within individual sheaths. In support of this model, purified myelin contains a free and polyribosome ribosome fraction [5], electron micrographs revealed ribosomes in the distal ends of oligodendrocyte processes [70], and newly synthesized MBP reporter proteins were visualized in cultured oligodendrocyte precursor cells [10]. Furthermore, our own work indicates that Fmrp [45] protein is localized to nascent sheaths. This evidence raises the possibility that local protein synthesis of translational regulators may modulate localized protein expression within myelin sheaths. Testing this model directly will require methods for visualization of de novo translation in vivo [71–75].

The candidate 3′ UTRs we selected were isolated from genes encoding cytosolic and transmembrane proteins. Canonically, transmembrane proteins are translated in the rough endoplasmic reticulum, processed by the Golgi apparatus, and transported via the secretory pathway. Identifying mRNAs encoding transmembrane proteins in the myelin transcriptome suggests that noncanonical pathways regulate protein synthesis of transmembrane proteins in nascent sheaths. Consistent with this hypothesis, we found that the 3′ UTR of *lrrtm1*, a transcript encoding a transmembrane protein, is sufficient to localize mRNA to nascent sheaths. In contrast, we found that the *cadm1b* 3′ UTR is not sufficient to localize mRNA to myelin, although we have previously shown that the Cadm1b protein is present in myelin and regulates sheath length and number [42]. Together, these observations raise the possibility that some transmembrane proteins are locally synthesized similar to dendrites [14,76]. In support of this hypothesis, distal oligodendrocyte processes contain satellite structures of rough endoplasmic reticulum [77]. Future work will need to test the hypothesis that some transmembrane proteins undergo noncanonical synthesis near sites of sheath growth.

We identified *cis*-regulatory elements in the 3′ UTRs of myelin-localized mRNAs. To begin, we bioinformatically identified 3 motifs common to the *mbpa*, *eif4ebp2*, and *fmr1* 3′ UTRs. Each motif was necessary for localization, but only 4 sequences were sufficient to drive mRNA to nascent sheaths. However, we are not able to exclude the possibility the motif 1 in the *mbpa* 3′ UTR contributes to mRNA localization. We interpret these data to mean that *mbpa* motif 3, *eif4ebp2* motif 1, *eif4ebp2* motif 3, and *fmr1* motif 3 are minimal *cis*-elements that regulate mRNA localization to nascent sheaths in vivo. The 4 other primary sequences isolated from the candidate 3′ UTRs, while necessary for localization, were not sufficient and thus require additional localization signals present in the 3′ UTR, which are currently unknown.

The capacity for motifs to promote mRNA localization when isolated from specific 3′ UTRs can be explained by several alternative possibilities. First, the primary sequences in the motifs

are important for interacting with key *trans*-acting factors, such as RNA binding proteins, that are only able to interact with that primary sequence. Second, primary sequences may form a secondary structure that is sufficient for myelin localization. Third, RNA modifications to the sequences may contribute to the variability in mRNA localization. Importantly, these alternative explanations may not be mutually exclusive and together demonstrate the complexities underlying mRNA sorting.

We found that motif 2 is enriched in the myelin transcriptome and approximately two-thirds of these motifs are positioned in the 3′ UTRs, suggesting that motif 2 significantly contributes to mRNA localization in oligodendrocyte. To the best of our knowledge, motif 2 does not correspond to any known mRNA localization signals in oligodendrocytes. GO analysis of myelin-localized transcripts containing motif 2 indicates that many of these transcripts encode proteins with biological functions important in cellular and neural projections. Specifically, we identified genes associated with neuronal cellular compartments such as axon, dendrites, synapse, postsynaptic density, neuron projection, cell projection, synaptic vesicle membrane, and terminal bouton. These data support the model that myelin sheath growth utilizes molecular mechanisms similar to synaptogenesis [42]. Together, these data raise the possibility that motif 2 mediates localization of a cohort of mRNAs that encode proteins that function in distal ends of cellular projections, implicating the motif as a mRNA regulon. RNA regulons are primary sequences or secondary structures that are co-regulated at the posttranscriptional level to coordinate cellular functions [78–81]. Identification of the *trans*-acting factors that interact with the motifs we identified will provide future insights into the molecular functions of myelin sheath growth, which will lead to a more complete understanding of the mechanisms guiding developmental myelination.

## Materials and methods

### Contact for reagent and resource sharing

Further information and requests for resources and reagents should be directed to and will be fulfilled by the Lead Contact, Bruce Appel (bruce.appel@ucdenver.edu).

### Ethics statement

All procedures were approved by the University of Colorado Anschutz Medical Campus Institutional Animal Care and Use Committee (IACUC) and performed to their standards.

## Experimental model and subject details

### Zebrafish lines and husbandry

All nontransgenic embryos were obtained from pairwise crosses of male and females from the AB strain. Embryos were raised at 28.5°C in E3 media (5 mM NaCl, 0.17 mM KCl, 0.33 mM CaCl, 0.33 mM $MgSO_4$ (pH 7.4), with sodium bicarbonate) and sorted for good health and normal developmental patterns. Developmental stages are described in the results section for individual experiments.

The transgenic line *Tg(mbpa:EGFP-CAAX-polyA-CG2)*[co34] was created by Dr. Jacob Hines. The transgenic construct was created using Gateway tol2 kit (Kwan and colleagues, 2007). Specifically, p5E-*mbpa* contains 2.6-kb genomic fragment of zebrafish *mbpa* (Hines and colleagues, 2015). *pME-EGFP-CAAX*, *p3E-polyA*, and *pDEST-tol2-CG2* were created by Dr. Jacob Hines. All entry vectors and destination were combined using LR clonase and transformed into DH5α cells. Colonies were screened by enzymatic digestion using BamHI, KpnI, and XhoI. Plasmid DNA was injected into AB embryos which were screened for transgenesis and

outcrossed to create transgenic lines. All *Tg(mbpa:EGFP-CAAX-polyA-CG2)$^{co34}$* used in this manuscript were from F3 or later generations.

## Method details

### Candidate 3′ UTR selection

To select 3′ UTR candidates for cloning into the MS2 system, we utilized published transcriptomics data [27]. We downloaded S1 Table containing transcript abundance in 4 stages of myelin development identified by RNA-seq. We selected the 3 biological replicates from P18 for analysis because this developmental time point was the most similar to our model. We filtered these data for transcripts with normalized read counts greater than 20 for all 3 biological replicates (representing 21,937 genes). We put all gene names into a GO analysis (geneontology.org) and analyzed the genes for biological processes in Mus musculus. From these biological processes, we copied all genes into an Excel document that fit the term "synap," "translation," "myelin," and "oligodend." Biological terms identified in GO analysis are listed in Table 1.

After removing duplicate genes with a GO term, the "Synap" list contained 855 genes, the "Translation" list contained 534 genes, the "myelin" list contained 128 genes, and the "oligodend" list contained 28 genes. To further narrow our search, we cross-referenced these lists with one another to find genes that were common to more than 1 list, which resulted in 55 genes. To further narrow this list, we cross-referenced these genes with the Brain RNA Seq online database [44] to identify those with evidence of oligodendrocyte lineage cell expression. We next referenced these genes with the zebrafish genome browser (GRCz11) and searched for annotated 3′ UTRs for each. Finally, we performed literature searches for published data that were relevant to our model. This resulted in a final list of 10 candidate 3′ UTRs: *dlg1*, *cyfip1*, *eif4ebp2*, *fmr1*, *cadm1*, *lrrtm1*, *eif4g1*, *eif4a3*, *mtmr2*, and *nfasc*.

### 3′ UTR cloning

To clone the *mbpa 3′ UTR*, 5-dpf cDNA from zebrafish larvae was used for PCR amplification using primers to target the flanking regions of the *mbpa* 3′ UTR. The PCR fragment was cloned into pCR2.1 TOPO using the TOPO cloning kit. Colonies were screened by colony PCR. Using Gateway cloning, the *mbpa 3′ UTR* was amplified and inserted into pDONR-P2R-P3 using BP clonase. *p3E-mbpa 3′ UTR* was confirmed by sequencing. All cloning steps were performed by Dr. Jacob Hines.

To clone the additional full-length 3′ UTRs, cDNA was created from pooled 6-dpf AB larvae treated with 1 mL of Trizol and snap frozen. All RNA isolation steps were performed on

**Table 1. List of biological process for GO terms.**

| "Synap" | "Translation" | "Myelin" | "Oligodend" |
|---|---|---|---|
| Regulation of Synaptic vesicle cycle | Translation | Regulation of Myelination | Oligodendrocyte Differentiation |
| Regulation of trans-synaptic signaling | Positive Regulation of Translation | Negative Regulation of Myelination | - |
| Synaptic Signaling | Regulation of Translation | Ensheathment of Neurons | - |
| Synaptic Plasticity | Negative Regulation of Translation | Paranode Assembly | - |
| Synaptic Vesicle Cycle | - | Myelin Assembly | - |
| Synaptic Vesicle Localization | - | Central Nervous System Myelination | - |
| Synapse Organization | - | - | - |
| Positive Regulation of Synaptic Transmission | - | - | - |
| Regulation of Synapse Structure or Activity | - | - | - |

ice and in a 4° centrifuge at 18,078 × g. Larvae were thawed on ice and homogenized with a 23-g needle. A volume of 200 μL of chloroform was added and shaken for 15 s followed by centrifugation for 10 min. The aqueous layer was transferred to a new tube, and an equal volume of 100% cold isopropanol and 2 μL of glycogen blue was added to the sample. The tube was incubated at −20° for 20 min and centrifuged for 10 min. The supernatant was removed and transferred to a new tube, and 200 μL of cold 70% ethanol was added to wash the pellet followed by 5-min centrifugation. This step was repeated. After the pellet dried, the RNA was resuspended in 20 μL of molecular grade water. RNA was quantified using a Nanodrop. To synthesize cDNA, we followed manufacturer instructions from the iScript Reverse Transcription Supermix for RT-qPCR, which uses random hexamer primers to synthesize cDNA.

To amplify the 3′ UTRs from cDNA, we designed primers that flanked the annotated 3′ UTR as predicted by *Danio rerio* GRCz11 annotated genome. Primers were flanked with attB sequences (Table 2) for cloning into the pDONR-P2R-P3 vector of the Tol2 Gateway kit (Kwan and colleagues, 2007). cDNA was used as a PCR template to amplify the 3′ UTRs. Of the ten 3′ UTRs we attempted to amplify, we were successful with the 6 listed below. Following amplification, bands were gel extracted using a Qiagen Gel Extraction Kit and cloned into pDONR-P2R-P3 using BP clonase. Clones were verified by sequencing using M13 forward and M13 reverse primers. The p3E-*dlg1* 3′ UTR was not fully sequenced due to highly repetitive sequences. We sequenced approximately 51% of the construct from 1 to 54 and 775 to 1,552 base pairs. We therefore confirmed p3E-*dlg1* 3′ UTR identity using restriction enzyme mapping.

The *sv40* 3′ UTR is a transcription termination and polyadenylation signal sequence isolated from Simian virus 40. We obtained this sequence from the Tol2 Gateway-compatible kit where it is referred to as "pA." This sequence was cloned with Gateway BP clonase into pDONR-P2R-P3. The p3E-sv40 3′ UTR was confirmed by sequencing.

## MS2 plasmid construction

All MS2 constructs were created using Gateway cloning. pME-*HA-NLS-tdMCP-EGFP* and pME-*24xMBS* were generous gifts from Dr. Florence Marlow.

To create pME-*mScarlet-CAAX-24xMBS*, we obtained plasmid pmScarlet_C1 from Addgene. In-Fusion cloning was used to assemble mScarlet-CAAX in puc19. Next, we amplified mScarlet-CAAX sequence using primers 5′-ggggacaagtttgtacaaaaaagcaggcttaatggtgagca agggcgag-3′ and 5′-ggggaccactttgtacaagaaagctgggtttcaggagagcacacacttgcag-3′ and cloned it in

**Table 2. Primers for UTR amplification and cloning.**

| 3′ UTR name | Forward Primer (5′->3′) | Reverse Primer (5′->3′) | 3′ UTR Length (nt) | Percentage of annotated 3′ UTR cloned |
|---|---|---|---|---|
| *lrrtm1-201* | ggggacagctttcttgtacaaagtggtatccacccatgtcagtttttacaaatcaatg | ggggacaactttgtataataaagttgttgttttccacttcaattgtgtctgttcg | 368 | 93.6% |
| *fmr1-201* | ggggacagctttcttgtacaaagtggtaccttcccctcattctcccact | ggggacaactttgtataataaagttgtttgcagaggaagatcaacctttatttattgaaa | 2,486 | 99.4% |
| *eif4ebp2-201* | ggggacagctttcttgtacaaagtggtaagaagaggaacctacgtgaacaac | ggggacaactttgtataataaagttgtgtccactggcattggca | 2,282 | 98.5% |
| *dlg1-201* | ggggacagctttcttgtacaaagtggtaggggccgaagacaaataaacct | ggggacaactttgtataataaagttgtatggaatgaatcaagttggcagattatgtac | 1,522 | 97.2% |
| *cyfip1-201* | ggggacagctttcttgtacaaagtggtagcaccagtttgaagtggaagagat | ggggacaactttgtataataaagttgtaaaaaggcacgtttatgaggagtaagaac | 585 | 88.1% |
| *cadm1b-201* | ggggacagctttcttgtacaaagtggtactggaactagacctgttagcttcc | ggggacaactttgtataataaagttgtcattttaaactgcttttattcactgttataatt | 1,220 | 100% |
| *mbpa-201* | gccttctccaagcaggaaaacactgagatg | gcagagtatatgagacacagaac | 1,174 | 86% |

plasmid pDONR-221 using BP clonase to create pME-mScarlet-CAAX. Next, we designed primers flanked with BamHI cut sites (5′ -tccggatccatggtgagcaagggcgaggcag-3′ and (5′-cgactc-tagaggatcgaaagctgggtcgaattcgcc-3′), and PCR amplified the mScarlet-CAAX sequence. We purified the amplified product using QIAquick PCR Purification Kit and digested it with Bam-HI-HF. pME-*24xMBS* was linearized with BamHI-HF and treated with Antarctic phosphatase to prevent religation. We performed the ligation with 2X Quick Ligase, and the ligation reaction was transformed into DH5α competent cells. Clones were screened using restriction mapping, then sequenced for confirmation.

For expression plasmids containing full-length 3′ UTRs, we used Gateway multisite LR clonase to combine entry vectors with pDEST-tol2. The resulting expression plasmids included: pEXPR-*mbp:mScarlet-Caax-24xMBS-mbpa 3′ UTR-tol2*, pEXPR-*mbpa:mScarlet-Caax-24xMBS-lrrtm1 3′ UTR-tol2*, pEXPR-*mbpa:mScarlet-Caax-24xMBS-fmr1 3′ UTR-tol2*, pEXPR-*mbpa*:mScarlet-Caax-24xMBS-eif4ebp2 3′ UTR-tol2*, pEXPR-*mbpa:mScarlet-Caax-24xMBS-dlg1 3′ UTR-tol2*, pEXPR-*mbpa:mScarlet-Caax-24xMBS-cyfip1 3′ UTR-tol2*, pEXPR-*mbpa*:mScarlet-Caax-24xMBS-sv40 3′ UTR-tol2*, and pEXPR-*mbpa:mScarlet-Caax-24xMBS-cadm1b 3′ UTR-tol2*. LR clonase reactions were transformed into Stellar Competent Cells (Takara cat # 636763). Clones were screened using restriction mapping.

To delete motifs, we used New England Biolabs Q5 Site Directed mutagenesis kit. Specifically, we designed primers flanking the motifs to omit the localization sequences from p3E-full length templates. We followed instructions outline in the kit to generate specific deletions. This step was repeated sequentially to delete all motifs from the previous template (Table 3).

To insert motifs into a Gateway entry vector, we provided Genscript with Gateway entry vector pDONR-P2R-P3 and the sequences for each gene to be synthesized. Genscript synthesized the sequences and cloned them into the Gateway entry vector between the attR2 and attL3 sites. Motifs (underlined) were separated by 3 to 4 random nucleotides.

The sequences synthesized from *fmr1* motifs were 5′- ggagctctagtctcccactcagctcgcacaccagtcgataaaaaaaaaaaaaaaaaccttttgaagtgattc-3′, from *eif4ebp2* motifs were 5′- ggagctctagatcaaaggtaggaaagcactgaagtggcactccagtcgattctcacagtgatctcattc-3′, from *mbpa* motifs were 5′-ggagctctagcaccagtcgctcaaaaaaaaggaaggaaaacctgaaagtggatattc-3′.

For control experiments to determine the specificity of mRNA detection by MCP-EGFP, we created an expression plasmid that lacks the 24xMBS stem loops (pEXPR-*mbpa:mScarlet-Caax-mbpa 3′ UTR-tol2*).

## Lifeact cloning for F-actin reporter

The F-actin reporter was created using Gateway cloning. Alexandria Hughes created pME-life-act-mNeonGreen by PCR amplification using primers 5′- ggggacaagtttgtacaaaaaagcaggctacca

**Table 3. Primers for motif deletions from full-length 3′ UTR.**

| 3′ UTR name / motif ID | Forward Primer (5′->3′) | Reverse Primer (5′->3′) |
|---|---|---|
| *mbpa* / motif 2 | caagatggataatgtgggg | tcatacccttttccttttatg |
| *mbpa* / motif 3 | gcagcgagtttaacagac | agtctgtagggcagacatc |
| *eif4ebp2* / motif 1 | ttgtgcttagcctccgta | ggaaaaaataaatcattctgtgcc |
| *eif4ebp2* / motif 2 | ttgtttttggtcatcgtac | ttgattatcaaggttcgtg |
| *eif4ebp2* / motif 3 | tttgtctgaggcacagaatg | cacagaaaaagacaattaaagtc |
| *fmr1* / motif 1 | caccaatccagatgcttc | atgagggggaaggtaccac |
| *fmr1* / motif 2 | ttgccaaacagactgtttc | ttttaacagactggtgaac |
| *fmr1* / motif 3 | tgttaaaaaaaaaaaaaaaaaaaacctttgaag | aactctgccatcttgcca |

tgggcgtggccgacttga-3′ and 5′- ggggaccactttgtacaagaaagctgggttcttgtacagctcgtccatgccca-3′ from mNeonGreen-Lifeact-7. We then combined entry vectors and pDEST-tol2 using LR clonase to create pEXPR-*mbpa*:*lifeact-mNeonGreen-polyA-tol2*.

## 3′ UTR sequences used for MS2 RNA localization experiments

Underlined sequences indicate the sequence motifs deleted or inserted in motif experiments.

*sv40* 3′ UTR

5′-gatccagacatgataagatacattgatgagtttggacaaaccacaactagaatgcagtgaaaaaaatgctttatttgtgaaatttgt-gatgctattgctttatttgtaaccattataagctgcaataaacaagttaacaacaacaattgcattcattttatgtttcaggttcagggg-gaggtgtgggaggtttttt-3′

*mbpa* 3′ UTR

5′-cttctccaagcaggaaaacactgagatggaagagagtgaaatggacggaaagcaaaaacttgagagggaggatgtctgccct acagact<u>caccagtc</u>gcagcgagtttaacagactaacattggccatcttcgcttcctagatagagatacaatccaagtatctgttgctacat gcctgcagggttacagaagcacgtgttgactgtatgtgtgcaaacttgctgtaataattgtcaatggtcaggtgatgcgatacatcttgtaagt ctccctttaaaatttagctgaagtgatcaatttcaatatatacaaagagcaaaaactcatcaaaaggtttgaaacaattagacagagtattt ctctttttttaaaatccctgaaccaaccagatgaatcatttgatcattctgaattggtcttatatgtgtttcacaaaatggattgcttatatgctct ccagcatttgatgtgtggcatttattctatgttatactgcctctccatggtttcttgagatccatgttcaacctcatgtgatgtgcatttctgtat gtttgtgttcactgtggtctttgtgttgcattctatattggttatttactttataccaggaatttgtatataggaatctttattgagtttaattaat gcaaaaaaaatatgatgaccagttaactacattaaatttgattcattttgagaaatgatttagctcttaatcaggaccatgccttaaaatgatt aaaaacagactaaaaacacaattttgtggactagggaatagattctaagcatgtatgtggcttggattgtatgccctcagatgttgcact gcagtatgtgtgttaaaaccacctgtaaatgttgtctgcgtcattacatgtgcaattttggtgttattttagaaaggtctcgtaattaccaggggt aggattaatatttaaataacaaacatgcagaaatctaggacaaagagtcagtgggagcataaaggaaagggtatg<u>aaaaaaaggaagg aaaacctgaaagtg</u>caagatggataatgtgggaaatgctaaatgaggacttctgaaagagtaaggtggagtttattcagctgattttttttt ctttttctgtgttgtatctcatgtgtcttaatatcgttcattgttctgtgtctcatatactctgcc-3′

*eif4ebp2* 3′ UTR

5′-aagaagaggaacctacgtgaacaacgattaattacctggtacctgtgtgccagtggcttggcttgtagataccaatgttgt gagccctctcctttagctctctctagctgctgggtgctgtttaatcatggggat+aaatgactaaagtttgcccagtggtgttgctggagccct gaaagttaacctgtgagccctgttgagctcttctttttttgttgtttaggtttcagggctgccatcagcagtacttgcttgagtcacagagcaa gggaaaaattctctattgcaagtgcccggatatattaccaatttaccataatagactcataaacggatcagccattagagattcactgctgat agatcaaagtacactgttccagttgatgcccttaatgcagtctatttcgttcacacataaactgatttgcaggacatgcggttatcattatct ccctatattttcgattgttttttcccccagttaattttaaaagggacaacaggattatattcatatttttttgattcaaaaagtgaaaaaattccaat cattcctgctcttttatatcattttttttttttcacctcaatacattttcaggtgttcaaaagaatagtttttatattgttgtccaaaaatgcataatt aaaccattgaccttaatcattcatttgaactgacacatgatttcaatgccaaatgtgcacagaatttctcattttatttaggactacccccaaaat aagcatttcaggtcatattagtttaatttcccagttttttcactgttaattttaagaaaagaaacacctaatcacattcgtgaaagaccaaact gttactttagattaaagagaatgtttctgttcagatttattttgctaacagtaaggtttgtttctttgtttgtctttgagagatgaagcgtcaacat agcctagttaaaagttaataacaccaagtaatgtttgattggataaagggtttaatataccttataagatgttgacaaagaaagctggtct caggaagcacttgctttatgcacctaacaattacgcatcttacggctcctgttacgggagaaaaagcacggtgaatatttaacattaatat cagctgtgacatctgctgttactttccgaaatactacaatcctcacgaaccttgataatcaa<u>atcaaaggtaggaaagcactgaagtg</u>tt gttttggtcatcgtaccatcggtctgtctcataagcgcacacaatgtgaacgagtttgtcgcgagcagacattcattttgattcagat caaacgtctcgaaaacttgaattgaatctttccaatgctcttctatgcatgttctttcggtctgcttcttaagctttagttcacatgtcagat cattttaacgttgttgttttttgggggggggacgccaaatcgtgtggaagacgagaattattattgtaagagaattaagcaaagagattatt caaacaaagtgacttaaagaggataagacttattctcatgactttaattgtctttttctgtg<u>ctccagtc</u>tttgtctgaggcacagaatgattt atttttcct<u>ctcacagtgatctc</u>ttgtgcttagcctccgtatccatgccttactttactaagctgagcagctgtgtcaaatacagtaccctttcat accttaaaacgggatcagttcctttctttcgtttctttcaagtggagaacaagatcaagatattgctgtttttactagaaaacatttatattttt ggagacatgtttgttatcatttactttcttacagaaatgggacgatttggaaatgcagaggtgtaatatttgtaatagtaatggattggtt aaaagcaaatacagaatttaggtttgactttgaggtcacagttaagtttctattcaattttgagaacttgtagaatccttggagtatgttttcatgt ccggtttgcgcaaaagtcacattcaagttctaatgatttacagtgaagggaaattggtgtatactgtccttctgtaatggatatgaatggt caaattccagcttgtttcaactctaaaattatgaaatgttaaattttttttatttaaaaaaaaaataatgcttactattgctttatgcttt

ccccacgcatcaaactatcagaaatgcatttagttctgtgtgaggggatttaagatggacgtgtttttatctaatatggcaaaaaacatt
ggaaactgtattttctttcttctttgaccctgttaattttaatgaaatgccaatgccagtggacac-3′

*lrrtm1* 3′ UTR

5′-tccacccatgtcagtttttacaaatcaatgtacgggtggatatggaacatacgttaacttggcacccaattttgctgctctcaaaggg
agctacagttctggagtgtgagtggactacaaacatggatttagagctgatttcaacagcctcatggggaaatctgactgtgagacccgt
gcacttgatgcaaaagatgtggaatgattatgctgaccagtcctggcttctcttgtgaaaagtggatatttgagctttaacgtgtctttctactt
caggatattctaggactctctaaagctaccgaggacatcaagtacaccatggtaacaaacatacgaacagacacaattgaagt
ggaaaacaacaa-3'

*fmr1* 3′ UTR

5′-ccttcccctcatt<u>ctcccactcagctc</u>caccaatccagatgcttcttcattagagacacattaggccaaagagaaccaggtcagt
agggctgtcgcaagacatgacataaagcacactttgtaattgttagcggggttaaaagacaaggtctcttggtgtacggtgtgttgaattt
ggtagtcttcacggattagagagaccggtctctaaatctcaactgaagctacaggttttctaattgactttctaaataactacctaaagaagct
gtggataaaatgtctctcataaatggaataaaaacaaaaaaaagagctttttccaagaaattggtttttaaagtttgtttttccacaatgt
ccaaaaaaaagggaaagaaacgcatcggaggatgaactcaagctctcgacgcccaagtttatgccacatctcatggcatacattgttttt
acacgctttggtgtccatggtttctcgagcccattctgatgtgagatcttaaccacatcattagcaaaaaaatacaaaaaaaatttaaaaaac
gaagagaaaaaataagtaaaaaataaaaaaaatgctcctgtctggatctgtgcctttaactagtgtggatagtatcataatttggatgttt
aaatcagggaagaaggcgtgaacaacacctgtcacctgagcagagtttgttgcaatctcagccgacaactttaaaaacaaacaaa
aaaaggtgtttcttttttttttctctgtgctgcatgtatttcacaaacattttgtatgtctgaaaagcaggattggaaatctattttgccatagct
gagttgtgagtagtaaaggtattagcctaactaaactagtcaactgtggaagtcattgattgtaactaaccacccgagttgaatatcaatgct
aattcaacaatgaaagcattaatgaagattaatttgatgtaaagcatatctttgcattttagaacaattgtttttttgtttttcaatagaggt
agcaaagcagtttcctttaaacaaaatgtctttttgtttttttacttgttttttacctccatcatgatataaaggtgaatttgctcagatgtttttct
cacaaacagggtggaaatgctctattttgtttcattttttttttttttgttcatgctggccacagtaaactgccatctttcctgctgtattttttcagcat
caggtgtgaagcacttatatcagattatatctgtaacattgtctttgtcctaacaatggctcatcatagactgaaaatgttgaactgtgggatt
gcatactgtatattattacatcatccgttaagagattgttgctgttcttttttagattgaactttaaatgtgcacttatcatgtttgtgttactct
gcaagtactttataatctataaacctataataaactagaatgtgataagacttcttcagcaggtgaatcacatgtatgctgtcaacataact
aaactgaagtttagatctcttcgctaattttgagtctactgtactttgtcagcttcgagagggacgagatttgggggaaacacaacgttaacttt
gatatgaggagaacaaaatgtctatcttatccgattagtgcttgtgcattaaactttagtttacaattttgagaagacgcaaagctgttct
aaacccacgattctttgaaacctactgtgacggtaaacatggtagtttttttgactgtacagcaagaaagccagtcagttgtatcctctttgct
ccttttttcttaacaactcttttcattttatgctcgctagtcaaatatccagctctacctgtaaagcttacaccagtttcaaagtgattttgcctt
gatcagtgtcctgtgaaaaccaaacggaaggggttacgtttcacgcgaaatgcaaaaaaaacctgtgactggcagaggatttgcagctct
gtgggatgggaatgataaaccactcgaacggggcgtcctggctgcatccggcccccttttaaaatttagcctttatttggaagtaaggtgtt
aaaatggtttactcccagtattttaaaaaaacttgcacatctacggtgtgcagtaaactggcaagatggcagagtt<u>caccagtc</u>tgtt
aaaa<u>aaaaaaaaaaaaaaaacccttt</u>gaa<u>gt</u>gttgccaaacagactgttttctagttatttattttttggaatgtatatgaaaaatgacaaattgt
aaaaccatctcttgcacacatcgttaggctattgtgatttgaataaagagctgaaaaggaataaaaacaaacaaaaaaacatgaaatactgt
agatactgaaccgaggtagactgaccttgtatgttactgcactttgggtgataatttctttgtacataatagcagaacagactgggtttatgtt
gacattgtttggttatagtgcaatatattttgtatgcaagcagtttcaataaataaaggttgatcttcctctgcaa-3′

*dlg1* 3′ UTR

Note: dlg1 3′ UTR contains a region of thymidine repeats that inhibited sequencing a 35 bp region (nucleotides 7 to 41). Within the 35 bp region, 5 nucleotides were not able to be confirmed and are underlined in the sequence below. Together, we confirmed 99.7% of the dlg1 3′ UTR sequence used in experimental procedures.

5′-ggggccgaaga<u>c</u>aaat<u>a</u>aa<u>c</u>cttacacttctttt<u>a</u>acttttt gtatttttttttaatcttttcgtttcctttttttaatattaacatggcct
gcagcttgcattggctttccaactcctgagcataagaaatgcgtttctttgaatgggtttggggttttttcctctttctctcatgcctctctttt
gaaactactgtctagaactcgcatcccaatctccgcatgtggtcctccactgggaggagacgtgcttgaccagcgacttactggacagatt
atacggtacagattccttaatgtttaaggggaggtgactttctcgaaggaaaaccaataagttaatgcattatacacattttggtttgttctt
cattttttacccccccagtatatgatcagaagctttcatgtctgtctctggaaagacataaagcaaactttgcacgtttttttgcatgtttggatt
attctgtttcagttaacatattgttcttacgtttttatgtcagacagatttaaaccatgtgactttcagccaccatatgactttagtttttcttct
aaaatcaagcaacccgttcacttt atgcagctgatctgttttt gtcgccagtacacaagaggatgataaaaaatttaatatttggcacaaa
agttcttcttttt atattaaatgtctatttgtcttaacgatctgtgtacattttaattaaggtgcatctgggagtagcttttagcagaggaacg

ggcttacgattatatttcaaacccagggtgacgacatattcttacttcaaaaaaacagcaatgcaaataagagagtataaattgtaatgtat
attcaaccctcagtatactgcctattttgttataatgaaagtcctatattgagctttaattaaactctacacctatgggaattattttctttataatgt
caagcatacactcgaatgactatgcgtgtatgtatataaaataaatataaatagatatatattttttcctttacttttaccaccatcactttttgtt
gtttctggtttgtctttatgcaaaaacactgacacacacacgcacactaggctggatggtggatttccatttgcatctgctccaacaatat
gaaaacaaaactcaagttggcagtttgtagtattcagctctctctttgtctctctctcctgtgtaattctgtcatattttctttctcatgtgacgt
gcatgatttcttaaagcaatagtcttcatcagcaaggagcggaggagagagggatttggcataaactgtaaacctgaaaggctgtatgggaatt
gttgaatgcccgtagctgaagagagatttgctttgcctttaactgtccaccagggggtgctgacggcatccccgctattttgttttcttt
cgttttttaactcggagatgattgttttctaaaagattgcttctctaacttgtcatcatggctttctaacacattttgactagataatgtacataat
ctgccaacttgattcattccat-3′

*cyfip1* 3′ UTR

5′-gcaccagtttgaagtggaagagatgggaaaagagagggaaatatttagcaacgtgtttaggaccagtttcactgtcacattctactt
aatgatgcctttcttatgccaccctgtagtttctgcagagccagtaagttgccttgtgattggtcgagtgatttgatttgcaccaattact
gcaccctactcaggctttggttaatcaggccaagatatgtttatgatgtacacaaacatggctcagttttactagcgtatttcaggtt
gatttttttgtacacataaacacacacatacacacacactcgtgccttattatagataaattgactataatctttatatgttaatgatagcac
agcactactcatgtttatacttgggaacagcagtggggtatggttccaaaatggccaagtgggttttatcaaagggttatttaagcttttgct
gtagtgcgagtccagcggtattttttctatgtggtgttgaatgattgcatgcaagttttttttttcttctattatacttaataaaactttaagcgt
agccaagttcttactcctcataaacgtgccttttt-3′

*cadm1b* 3′ UTR

5′-actggaactagacctgttagcttccagtgctgaacagcaaactgtggactgctgggtttgggaggaaggggttgtgggattcaat
caggctggattcacacctgcgcagctgaagagacgctgctgccatcgagacggacggggtgtgggatggcagagcactaggagagct
caattttagaccgcttcaccatccaacacctcctgctggggccggttttgttccgtttaaacatttgctaagaattttgtgccttgttctgttctt
actgaagttcccactttcatcaggacacccacaagctactgtctctccaatggtctgaatccacgtttttttttttctctcctttcatttcattttttt
aattgctagcacatcttaaagctctctctcatgctgttctgtctgcaggtctgaaagaaagccgcgcaccaagtttcgatgtaagattcagt
cacagaatgatcgcttgggttgttttcacggattgattccaagtaatttatttggcaaattgcctgttgctctccttagtcccccagagaacgag
ttagagatgatagaaacctttttttctctctcttttttgtgaccaacaacattcagcagcagtattgcattgttgcaatatttattggatatactgt
atgcgattatgatcagcttgtgttgatattagggctgtcaatcgaacacattactactattatttcgtttattaaatcattaaaatattacc
caaatttggcaatgacatgacaaaatttatattattattattattatatacagctatggaaaaaatattaagactaattatgacatttcttact
aaattaaaatgaaaaagttattatttagagcattgttttttttttttgcttgtttgtttttttctttaaaaaaaaaaggtgctaaattttcagacatgttgct
agccaaattcatttatatagaacatttcataaacagtaataattccaagtgctttacataaacaggaataaaagaaacaagtataagaaaat
aaaaaacaaattataacagtgaataaaaagcagtttaaaatg-3′

*mbpa* motif 2/3 insertions

5′-ggagctctag<u>caccagtcg</u>ctca<u>aaaaaaaggaaggaaaacctgaaagtgg</u>atattc-3'

*mbpa* motif 2 insertion

5′-ggagctctaggctca<u>aaaaaaaggaaggaaaacctgaaagtgg</u>atattc-3'

*mbpa* motif 3 insertion

5′-ggagctctag<u>caccagtcg</u>ctcagatattc-3′

*eif4ebp2* motif 1/2/3 insertions

5′-ggagctctag<u>atcaaaggtaggaaaagcactgaagtgg</u>c<u>actccagtcg</u>at<u>tctcacagtgatctc</u>attc-3′

*eif4ebp2* motif 1 insertion

5′-ggagctctaggatt<u>tctcacagtgatctc</u>attc-3′

*eif4ebp2* motif 2 insertion

5′-ggagctctag<u>atcaaaggtaggaaaagcactgaagtgg</u>caattc-3′

*eif4ebp2* motif 3 insertion

5′-ggagctctaggca<u>ctccagtcg</u>atattc-3′

*fmr1* motif 1/2/3 insertions

5′-ggagctctag<u>tctcccactcagctcgca</u><u>caccagtcg</u>at<u>aaaaaaaaaaaaaaaaacctttgaagtg</u>attc-3′

*fmr1* motif 1 insertion

5′-ggagctctag<u>tctcccactcagctcgc</u>aattc-3′

*fmr1* motif 2 insertion

5′-ggagctctaggata<u>aaaaaaaaaaaaaaaaacc</u>tttgaagtgattc-3′

*fmr1* motif 3 insertion

5′-ggagctctaggca<u>caccagtcg</u>atattc-3′

## smFISH probe design

The *EGFP* smFISH probes were purchased from Stellaris LGC Biosearch Technologies. Probes consist of a set of pooled oligos with CAL Fluor Red 610 Dye. smFISH probes were designed using the Stellaris RNA FISH Probe Designer tool by entering the zebrafish *mbpa*, *eif4ebp2*, or *fmr1* cDNA sequences obtained from Ensemble Genome Browser from transcript *mbpa-206*, *eif4ebp2-201*, and *fmr1-201* (GRCz11) (Tables 4–6). Probes with highly repetitive sequences were removed. Each probe was entered into BLAST to search for off targets and were removed if they were predicted to bind annotated genes with relatively high specificity. The probes were ordered with a CAL Fluor Red 610 Dye. Probes were resuspended in Tris-EDTA (pH 8.0) and stored at a stock concentration of 12.5 μM at −20˚C.

## smFISH experimental procedure

The smFISH protocol was adapted from 3 published protocols: Hauptmann and Gerster (2000), Lyubimova and colleagues (2013), and Oka and colleagues (2015). First, larvae were sorted for EGFP expression and fixed O/N in 4% paraformaldehyde at 4˚C. Larvae were embedded laterally in 1.5% agar and 5% sucrose blocks and transferred to a 30% sucrose solution O/N at 4˚C. Blocks were frozen on dry ice and sectioned with a Leica cryostat into 20-μm thick sections and placed on microscope slides. Slides were not allowed to dry more than 5 min before adding 4% paraformaldehyde to fix the tissue at RT for 10 to 20 min. The slides were quickly rinsed with 1X PBS twice. The tissue was permeabilized with 70% cold ethanol at −20˚C for 2 h. Parafilm was placed over tissue to prevent evaporation at all incubation steps. The tissue was rehydrated with wash buffer (10% DI formamide, 2X SSC in molecular grade water) for 5 min at RT. From this point on, care was taken to protect the tissue and probes from light as much as possible. Hybridization Buffer was made: 2x SSC, 10% DI formamide, 25 mg/mL tRNA, 50 mg/mL bovine serum albumin, 200 mM ribonucleoside vanadyl complex in DEPC water. Aliquots were made and stored at −20˚C. Final probe concentrations for egfp and mbpa was 125 nM. Final probe concentrations for eif4ebp2 and fmr1 was 250 nM. Slides were incubated at 37˚C overnight in probe. Slides were quickly rinsed with fresh wash buffer

**Table 4. Probe sequences for *mbpa-206*.**

| Probe Name | Sequence 5′->3′ | Probe Name | Sequence 5′->3′ |
| --- | --- | --- | --- |
| Probe 1 | ctttggattgagcggagaag | Probe 13 | aatcttcaacctgggagaaa |
| Probe 2 | gtccagactgtagaccactg | Probe 14 | gatctcgctctccacccaaa |
| Probe 3 | cagatcaacacctagaatgg | Probe 15 | ctggagcaccatcttctgag |
| Probe 4 | ctctggacaaaaccccttcg | Probe 16 | cttctccaagcaggaaaaca |
| Probe 5 | tgtcctggatcaaatcagca | Probe 17 | gagatggaagagagtgaaat |
| Probe 6 | ttcttcggaggagacaagaa | Probe 18 | cgggaagcaaaaacttgaga |
| Probe 7 | agagaccccaccactctt | Probe 19 | atgtctgccctacagactca |
| Probe 8 | tcgtgcatttcttcaggagc | Probe 20 | gtcgcagcgagtttaacaga |
| Probe 9 | tcgtgcatttcttcaggagc | Probe 21 | acattggccatcttcgcttc |
| Probe 10 | tcgaggtgggagagaactatt | Probe 22 | agatagagatacaatccaag |
| Probe 11 | cattagatcgccacagagac | Probe 23 | tctgttgctacatgcctgca |
| Probe 12 | aagggaacagaacacacttt | Probe 24 | ttacagaagcacgtgttgac |

**Table 5. Probe sequences for *eif4ebp2-201*.**

| Probe Name | Sequence 5′->3′ | Probe Name | Sequence 5′->3′ |
|---|---|---|---|
| Probe 1 | ggaaagcgataagaaacgag | Probe 16 | gcttggcttgtagataccaa |
| Probe 2 | gaaagggcccgaggtttta | Probe 17 | gccctctcctttagctctct |
| Probe 3 | ttccatcggggaaaacttat | Probe 18 | gctgggtgctgtttaatcat |
| Probe 4 | agcaagtgcaatgtcgtcca | Probe 19 | aaagtttgcccagtggtgtt |
| Probe 5 | cgtcagcttagtgagagcag | Probe 20 | gagccctgaaagttaacctg |
| Probe 6 | ctgatcaacgactcaacgca | Probe 21 | agccctgttgagctcttctt |
| Probe 7 | ctcacgactattgcaccact | Probe 22 | gcagtacttgcttgagtcac |
| Probe 8 | tggaggcactttattctcca | Probe 23 | agcaagggaaaaattctcta |
| Probe 9 | gaggaacccgaataatctat | Probe 24 | tgcccggatatattaccaat |
| Probe 10 | tcgtaagttcctgttggacc | Probe 25 | tgcccggatatattaccaat |
| Probe 11 | gaatgaaatcaagcggaatg | Probe 26 | tgatgcccttaatgcagtct |
| Probe 12 | catcaacaaccatgatgcca | Probe 27 | aaactgatttgcaggacatg |
| Probe 13 | caaggtgaagatgctcagtt | | |
| Probe 14 | agatggacatctaaagaaga | | |
| Probe 15 | aacctacgtgaacaacgatt | | |

followed by 2 wash steps at 37°C for 30 min. DAPI was added at 1:1,000 concentration in wash buffer to the tissue for 5 to 7 min at RT. Slides were quickly rinsed twice with wash buffer. Finally, slides were mounted with Vectashield mounting media and a No. 1 coverslip and sealed with nail polish. All slides were stored and protected from light at 4°C.

## Microscopy

To image RNA localization in living animals, plasmids were injected with mRNA encoding Tol2 transposase into newly fertilized eggs. Injection solutions contained 5 μL 0.4 M KCl, 250 ng Tol2 mRNA, and 125 ng pEXPR-*sox10*:*NLS-tdMCP-EGFP-sv40 3′ UTR-tol2* plasmid and

**Table 6. Probe sequences for *fmr1-201*.**

| Probe Name | Sequence 5′->3′ | Probe Name | Sequence 5′->3′ |
|---|---|---|---|
| Probe 1 | ggagctttctacaaggctta | Probe 18 | tgatctcgatgaagagacat |
| Probe 2 | cagccagaacgacagatttc | Probe 19 | tttcacatctatggagagga |
| Probe 3 | tccaggatgttcggtttcca | Probe 20 | ccgaagctacctggaatttt |
| Probe 4 | tccaaccggttttcagaaag | Probe 21 | aaaagtcattggcaagagtg |
| Probe 5 | taatgataaggaaccctgtt | Probe 22 | agctgattcaagaggttgtg |
| Probe 6 | caaagttcgcatggtgaaag | Probe 23 | taaatcgggtgttgtcagag |
| Probe 7 | acgttatagaatatgcagcc | Probe 24 | aaggagagcatttctaatgc |
| Probe 8 | tgatgccaccctaaatgaaa | Probe 25 | tattctgctggactaccatc |
| Probe 9 | gtcacattagagaggctacg | Probe 26 | tttaaaggaggtagatcagc |
| Probe 10 | agcaacaaagaacacctttc | Probe 27 | acccgagaaagaaaagtctt |
| Probe 11 | aaaaccagactagatgttcc | Probe 28 | caaacctttggtcgaggag |
| Probe 12 | agacttgagacagatgtgtg | Probe 29 | gagtctatgggttatcccaa |
| Probe 13 | ctctgaagaaaagcagttgg | Probe 30 | gatacaaaactgaggacatg |
| Probe 14 | atccatgttgagtgacatgc | Probe 31 | gtagttccagagactccaag |
| Probe 15 | tttaggagtctgcgcacaaa | Probe 32 | catcgacagcaataacgaga |
| Probe 16 | tcatgaacagtttgtggtgc | Probe 33 | gtagtgaacggcgtttcgta |
| Probe 17 | aagccagaaagatttctgga | | |

125 ng pEXPR-*mbpa:mScarlet-CAAX-various 3′ UTR-polyA-tol2*. Larvae were grown to 4 dpf and selected for good health and normal developmental patterns. Larvae were immobilized in 0.6% low-melt agarose with 0.06% tricaine. Images of single time point data were obtained using a Zeiss LSM 880 laser scanning confocal microscope equipped with a 40×, 1.3 NA oil immersion objective. Imaging was performed using Zeiss Zen Black software with the following parameters: 1024 × 1024 frame size, 1.03 μs pixel dwell time, 16-bit depth, 10% 488 laser power, 14% 561 laser power, 700 digital gain, 488 filter range 481 to 571, mScarlet filter range 605 to 695, and z intervals of 0.5 μm. All images of single cells were taken in the spinal cord of living zebrafish above the yolk extension. Cells were selected for imaging based on dual expression of EGFP and mScarlet-CAAX.

Images of smFISH experiments were obtained using a Zeiss LSM 880 with Airyscan confocal microscope and a Plan-Apochromat 63×, 1.4 NA oil immersion objective. The acquisition light path used Diode 405, Argon 488, HeNe 594 lasers, 405 beam splitter and 488/594 beam splitters, and Airyscan super resolution detector. Imaging was performed using Zeiss Zen Black software and parameters included: 1024 × 1024 frame size, 1.03 μs pixel dwell time, 16-bit depth, 3× zoom. Line averaging was set to 2, 2% to 5% 488 laser power, 2% 594 laser power, 0.5% to 3% 405 laser power, 750 gain, and z intervals of 0.3 μm. All images of single cells were taken in the hindbrain of zebrafish larvae. Cells were selected for imaging based on expression of EGFP-CAAX and Quasar-610 fluorescence. Post-image processing was performed using Airyscan Processing set to 6.8 for images that were quantified. For representative images of *fmr1* and *eif4ebp2* smFISH, post-acquisition processing was performed using auto Airyscan Processing.

## Quantification and statistical analysis

### Quantification of MS2 RNA localization

All images were processed and analyzed using ImageJ Fiji software. To analyze mRNA fluorescent intensity in sheath termini, we imaged single cells co-expressing NLS-MCP-EGFP and mScarlet-CAAX. Individual myelin sheaths were optically isolated by performing a maximum z projection of images collected at 0.5 μm intervals. Fluorescence intensity was measured by performing line scans across a 7-μm (± 0.3 μm) distance beginning at the terminal end of each sheath. Specifically, we drew each line in the mScarlet-CAAX channel to ensure we encompassed the edge of the myelin membrane. Gray values along each line (at 0.2 μm intervals) were measured in both channels. All measurements were combined into a Microsoft Excel file and imported into RStudio for further processing and analysis. In RStudio, we used tidyverse and ggplot2 libraries to manipulate data and generate plots. To normalize fluorescence intensities in each sheath, we divided the raw gray value at each distance by the average gray values of all distances per sheath. To calculate the average mRNA fluorescence intensity among all myelin sheaths, we plotted the average normalized fluorescence intensity by distance. To calculate mRNA fluorescence intensities in myelin sheaths, we plotted the average fluorescent intensity of EGFP (raw gray values) for each sheath using the line scan measurements described above.

To measure mRNA fluorescent intensity in cell bodies, we imaged single cells co-expressing NLS-MCP-EGFP and mScarlet-CAAX. Due to the high expression levels of EGFP in the nucleus, the 488 laser power was lowered to 0.3 to ensure we captured the full dynamic range of EGFP intensities. Cells containing saturated pixels were not utilized. During post-acquisition analysis, cell bodies were optically isolated by performing maximum z projection of images collected at 0.5 μm intervals. Fluorescence intensity was measured by drawing 3 regions of interest (ROIs) at the cell periphery, cell center, and between the periphery and center. Each ROI was 3 pixels by 3 pixels. All data points were combined in Microsoft Excel and

imported into RStudio for analysis. To calculate the average fluorescence intensity per cell, we averaged the 3 ROIs per cell. We normalized the intensity from each cell body to the average of all cell bodies in the *sv40* 3′ UTR control.

## smFISH quantification

All quantification was performed in ImageJ Fiji using a custom script created by Karlie Fedder (available upon request). First, z intervals were selected for individual cells or myelin tracts using the "Make Substack" feature in Fiji. Substacks for cell bodies included all z intervals for each soma. Substacks of myelin tracts in the hindbrain included 100 pixels × 100 pixels and 13 steps with an interval of 0.3 μm (4.39 μm × 4.39 μm × 3.9 μm). This volume was chosen because it was approximately the same volume as cell bodies. Each substack was maximum z-projected. Background was subtracted using a 2.5 rolling ball. The image was then thresholded by taking 3 standard deviations above the mean fluorescence intensity. Puncta were analyzed using the "Analyze Particles" feature with a size of 0.01 to Infinity and circularity of 0.00 to 1.00. Using the maximum projection of the EGFP-CAAX channel, an ROI was drawn around each cell body using the freehand tool. Alternatively, for myelin ROIs, the rectangle tool was used to draw a square 100 × 100 pixels (4.39 μm × 4.39 μm). All thresholded puncta were inspected to ensure single molecules were selected. Occasionally, threshold puncta fell on the border of the ROI, and these were excluded from measurements. *mbpa* transcripts are highly expressed, and counting individual puncta was not consistently reliable. Therefore, to measure each puncta, we overlaid the binary image on the maximum z-projected image and calculated the density (area × average fluorescence intensity) using the "IntDen" measurement.

To obtain the average mRNA abundance per subcellular compartment, we calculated the average density for all puncta in each ROI (cell bodies or myelin). All ROIs for each subcellular compartment were then averaged to calculate the average density per subcellular compartment.

To measure mRNA abundance and sheath length, we calculated the total density using the "IntDen" measurement in Fiji for each nascent sheath. Additionally, we measured the sheath length. All data were imported to Microsoft Excel and analyzed for statistical correlation in RStudio using Spearman's correlation coefficient.

## Statistics

All statistics were performed in RStudio (version 1.1.456) using devtools (version 2.2.1) and ggplot2(Wickham, 2009) packages. Additionally, several packages and libraries were installed including tidyverse (Wickham and colleagues, 2019), readxl (version 1.3.1), RColorBrewer (version 1.1–2), ggsignif (version 0.6.0), and ggpubr (version 0.2.4). All statistical analyses were performed with ggpubr. Wilcoxon rank sum was performed for unpaired comparisons of 2 groups.

## Bioinformatic analysis

### Identifying transcripts in the myelin transcriptome

To identify cDNA sequences in the myelin transcriptome associated with S1 Table in the main text, we started with the myelin transcriptome obtained from [27]. Specifically, we used the data from the 3 biological replicates from P18 mice, and these biological replicates were called "Treatment." As a control group, we used 6 RNA-seq datasets from 2 independent studies using cultured oligodendrocytes to eliminate any axonal-derived mRNAs [44,58]. Specifically, we used P7B2 and P7B3 datasets from [58] and the 2 NFO and 2 MO datasets from [44]. These

datasets were called "Control" in S1 Table. We calculated the average abundance (FPKM) in the treatment group and control group and determined the fold change. Next, we filtered the data for transcripts that have a q-value less than 0.05. Finally, to eliminate any genes with low mRNA abundance (FPKM), we filtered the data to only include genes that have FPKM greater than 5 in the control or treatment groups.

## MEME analysis

To identify motifs shared between the *mbpa-201*, *eif4ebp2-201*, and *fmr1-201* 3′ UTRs, we used MEME (version 5.1.1) part of the MEME suite software. We used the default settings: Classic mode, RNA sequences, Zero or One Occurrence Per Sequence, and set the maximum motif to identify at 20 motifs. We selected the top 3 motifs for experimental procedures.

## AME analysis

To identify if motifs enriched in the myelin transcriptome, we used Analysis of Motif Enrichment (AME) (version 5.1.1) part of MEME suite software. Specifically, we downloaded cDNA sequences in fasta formats for transcripts present in the myelin transcriptome (1,771 cDNA fasta sequences associated with the 1,855 genes in S1 Table). Some genes had multiple cDNA sequences that correlate to splice variants, and we selected the longest variant for our analysis. We uploaded these sequences into AME software and used the default settings to determine if motifs 1, 2, or 3 were enriched in the myelin transcriptome. As a control sequence, we used shuffled input sequences.

## FIMO analysis

To determine the frequency of motif 2 occurrences in the various datasets, we used Find Individual Motif Occurrences (FIMO) (version 5.1.1) part of the MEME suite software. Specifically, we downloaded cDNA sequences from the entire mouse transcriptome (mm10), cDNA sequences from the myelin transcriptome, 5′ UTR sequences from the myelin transcriptome, 3′ UTR sequences from the myelin transcriptome, or coding sequences from the myelin transcriptome. We uploaded the sequences in to FIMO software and used the default settings to determine the number of occurrences for motif 2. Table 7 indicates the number of input sequences for each condition, number of occurrences motif 2 was identified, and the number of unique genes with one or more copies of motif 2. The results from each of these analyses can be found in the supporting information tables.

## GO analysis

To identify GO terms associated with the myelin transcriptome and motif 2-containing myelin transcriptome, we used DAVID software (version 6.8). Specifically, we submitted Ensemble

**Table 7. FIMO analysis results.**

| Sequence Type | Number of Input Sequences | Number of Motif 2 Occurrences Found | Number of Unique Genes with Motif 2 |
|---|---|---|---|
| cDNA from mouse transcriptome | 56,289 | 36,662 | 16,144 |
| cDNA from myelin transcriptome | 1,771 | 2,101 | 751 |
| 5′ UTRs from myelin transcriptome | 1,195 | 115 | 59 |
| Coding sequences from myelin transcriptome | 1,411 | 470 | 751 |
| 3′ UTR from myelin transcriptome | 1,404 | 1,341 | 480 |

FIMO, Find Individual Motif Occurrences; UTR, untranslated region.

Gene IDs from the myelin transcriptome (2,821 genes) or the 751 genes containing motif 2. We selected GO term categories for biological processes, cellular compartment, and up_keywords. We filtered the results for false discovery rate (FDR) less than 0.05. We identified 60 terms in the myelin transcriptome and 34 terms in the motif 2-containing myelin transcriptome. We sorted the GO terms from lowest to highest FDR, removed any duplicate GO terms, and selected the top 20 terms.

## Supporting information

**S1 Table. Genes localized to myelin.** Control_mean is the average mRNA expression in oligodendrocytes across individual experimental runs. Treatment_mean is the average mRNA expression in purified myelin across individual experimental runs. GEO accession numbers are provided for individual experimental runs.
(XLS)

**S2 Table. Genes localized to myelin that contain one or more copies of motif 2.** FIMO, Find Individual Motif Occurrences.
(XLSX)

**S3 Table. Genes in mouse transcriptome that contain one or more copies of motif 2.** FIMO, Find Individual Motif Occurrences.
(XLSX)

**S4 Table. Genes localized to myelin and contain one or more copies of motif 2 in the 3′ UTR.** FIMO, Find Individual Motif Occurrences.
(XLSX)

**S5 Table. Genes localized to myelin and contain one or more copies of motif 2 in the 5′ UTR.** FIMO, Find Individual Motif Occurrences.
(XLSX)

**S6 Table. Genes localized to myelin and contain one or more copies of motif 2 in the coding region.** FIMO, Find Individual Motif Occurrences.
(XLSX)

**S7 Table. Gene ontology terms significantly enriched in the myelin-localized transcripts.**
(XLSX)

**S8 Table. Gene ontology terms significantly enriched in the myelin-localized transcripts that contain one or more copies of motif 2.**
(XLSX)

**S1 Data. Data analysis for Figs 1, 3, 4, 6 and 7.**
(RMD)

**S2 Data. Numerical data for Figs 1E, 3E, 3F and 4D.**
(CSV)

**S3 Data. Numerical data for Fig 1F.**
(XLSX)

**S4 Data. Numerical data for Figs 6 and 7.**
(CSV)

**S5 Data. Data analysis for Fig 2.**
(RMD)

**S6 Data. Numerical data for Fig 2C and 2D.**
(XLSX)

**S7 Data. Numerical data for Fig 2E.**
(XLSX)

**S8 Data. Data analysis for Fig 8.**
(RMD)

**S9 Data. Numerical data for Fig 8E.**
(XLSX)

**S10 Data. Numerical data for Fig 8F.**
(XLSX)

## Acknowledgments

We are grateful to Florence Marlow for her generous gift of the MS2 plasmids. We thank Karlie Fedder and Douglas Shepherd for their guidance during smFISH quantification.

## Author Contributions

**Conceptualization:** Katie M. Yergert, Bruce Appel.

**Formal analysis:** Katie M. Yergert.

**Funding acquisition:** Bruce Appel.

**Investigation:** Katie M. Yergert, Caleb A. Doll, Rebecca O'Rouke.

**Methodology:** Katie M. Yergert, Rebecca O'Rouke.

**Project administration:** Bruce Appel.

**Resources:** Jacob H. Hines, Bruce Appel.

**Supervision:** Bruce Appel.

**Writing – original draft:** Katie M. Yergert.

**Writing – review & editing:** Caleb A. Doll, Bruce Appel.

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
