## [Editor Report · Decision Letter 0]

11 Mar 2020

Dear Dr Appel, 

Thank you for submitting your manuscript entitled "Identification of 3’ UTR motifs required for mRNA localization to myelin sheaths in vivo" for consideration as a Research Article by PLOS Biology.

Your manuscript has now been evaluated by the PLOS Biology editorial staff as well as by an academic editor with relevant expertise and I am writing to let you know that we would like to send your submission out for external peer review.

Please re-submit your manuscript within two working days, i.e. by Mar 13 2020 11:59PM.

Kind regards,

Ines

--

Ines Alvarez-Garcia, PhD

Senior Editor

PLOS Biology

Carlyle House, Carlyle Road

Cambridge, CB4 3DN

+44 1223–442810

---

## [Decision Letter · Decision Letter 1]

24 Apr 2020

Dear Dr Appel,

Thank you very much for submitting your manuscript entitled "Identification of 3’ UTR motifs required for mRNA localization to myelin sheaths in vivo" for consideration as a Research Article at PLOS Biology. Thank you also for your patience as we completed our editorial process, and please accept my apologies for the delay in providing you with our decision. Your manuscript has been evaluated by the PLOS Biology editors, an Academic Editor with relevant expertise, and by four independent reviewers.

As you will see, all the reviewers find the results of your manuscript interesting and significant for the field. Nevertheless, they also raise several issues that need to be addressed, including some important controls. After discussing the reviews with the academic editor, we do feel that Reviewer 2's concerns about expression levels driving apparent changes in localization of transcripts, and Reviewer 3's requests for data formally linking mRNA abundance to final sheath length and better documentation of localization differences in Fig. 4 are essential.

In light of the reviews (attached below), we will not be able to accept the current version of the manuscript, but we would welcome re-submission of a revised version that takes into account the reviewers' comments. We cannot make any decision about publication until we have seen the revised manuscript and your response to the reviewers' comments. Your revised manuscript is also likely to be sent for further evaluation by the reviewers.

We expect to receive your revised manuscript within 3 months. We are aware that many labs are currently closed due to COVID-19 and that you might require an extension to perform the experiments needed. Please email us (plosbiology@plos.org) if you have any questions or concerns, or would like to request an extension.

At this stage, your manuscript remains formally under active consideration at our journal; please notify us by email if you do not intend to submit a revision so that we may end consideration of the manuscript at PLOS Biology.

**IMPORTANT - SUBMITTING YOUR REVISION**

*Re-submission Checklist*

*Published Peer Review*

*PLOS Data Policy*

*Blot and Gel Data Policy*

Sincerely,

Ines

--

Ines Alvarez-Garcia, PhD

Senior Editor

PLOS Biology

Carlyle House, Carlyle Road

Cambridge, CB4 3DN

+44 1223–442810

Reviewers’ comments

Rev. 1:

The manuscript "Identification of 3' UTR motifs required for mRNA localization to myelin sheaths in vivo" by Yergert et al., describes a method to visualise mRNA in living zebrafish larvae to study the mechanisms regulating localisation of specific mRNAs in myelin. It has been known for a very long time that one of the main protein constiuents of myelin, Myelin Basic Protein (MBP) is locally translated in the myelinating processes of oligodendrocytes, and a large number of stdies have been carried out aiming to understand how this occurs, but very few in vivo. Yergert et al., use the mRNA ecoding MBP as a starting point for their analysis and show very nicely that its 3'UTR signal is required to detect localisaiton to myelin sheaths using the technique that they employ for in vivo imaging. This validates the technique and provides an important platform through which to analyse the regulatory elements of other factos known to be localised to myelin sheaths. The authors identify regulatory motifs essential for the localisation of other mRNAs by bioinformatic data mining, and imaging using a series of well-controlled experimental and control constructs, and converge on the identification of a motif that is highly enriched across the "myeln transcriptome." It is now clear that a very large number of mRNAs are localised to the myelinating processes of oligodendrocytes and much remains to be discovered as to how these factors are transported to and translated at specific subcellular locations. This complexity is already indicated in the data presented in this manuscript showing that while necessary, one fo the key novel motifs identified is not sufficient for the localisation to myelin of fmr1 or eif4ebp2. The platform developed by Yergert et al., will continue to elucidate how mRNAs are localised in vivo, not only in myelin, but presumably also in synapses etc. The current manuscript, as it stands, is self-contained, the data are of a high standard, the presentation and text are very clear, and the adaptation of the technique and findings of broad interest. To me, this manuscript is already a strong candidate for publication in PLoS Biology.

Rev. 2:

The manuscript by Yergert et al. address an important and interesting research question namely how specific proteins are targeted to the growing myelin membrane. The mechanistic insight into how mRNA transport and local translation contribute to this is very limited. To identity novel mRNAs targeted to the myelin membrane and the cis regulatory element responsible for this process Yergert et al. used an elegant combination of in vivo (in zebrafish) and in silico approaches. The paper thereby provide important new knowledge that will help progress the basic understanding of myelin biology. The paper is well written and the data are organized in a logical manner. The findings are novel and of interest not only to myelin biologist, but also for cell biologist in general with an interest in understanding the mechanisms involved in regulating subcellular compartmentalization of proteins by mRNA transport and local translation. However, I have some concerns that need to be address prior to publication. These concerns are listed below.

Minor concerns

1) In the introduction on page 3 line 56 the authors state that 3'UTR of the beta-actin mRNA contains a sequence that is recognized by the RNA binding protein ZBP-1 for localization to dendrites. This zip- code is not exclusive for targeting to the dendrite, but are also used to target mRNA to axons (Willes et al EMBO J. 2011 and others) and to the leading edge of migrating cells. The paragraph should be rewritten to clarify this.

2) In the introduction the authors focus on the previous knowledge of the short sequence element previously identified as sufficient for localization of the MBP mRNA (RTS or A2RE element) to the process of mice oligodendrocytes in culture. In the paper they refer to by Ainger et al. JCB 1997 they also identify a region required for localization of the mRNA to the myelin sheath. It may be of interest also to mention this in the introduction. A comment on whether the identified motif 2 are found within this localization region in the mouse sequence would be of relevance for the discussion. Furthermore, a comment on whether motif 2 includes any recognition sequences for transaction factors previously know to be involved in transport or translational regulation of the MBP mRNA would be relevant to mention or discuss.

Major concerns

1) From the pictures presented in Figure 1 there seem to be a higher expression level from the construct including the 3'UTR of Mbpa. Can the authors exclude that the observed difference in localization is a result of higher expression from the reporter constructs in the cells expressing the variant with the 3'UTR. Comparison of cells with similar mRNA levels or expressing the about amount of sheath localized mRNA relative to total mRNA in the individual cell could be a possibility to overcome such problems.

2) In Figure 6 it is not entirely clear whether Mbpa full length in Figure 6E correspond to the construct present in Figure 6B and for which the above as missing the last app 180 bp of the 3'UTR and potential regulatory element 1 . If this is the case it would be more correct to label the construct 3-1183 of the 3'UTR as above and not full length. If the data presented are indeed from the truncated variant, the authors should include data on the full-length 3'UTR construct to exclude any positive or negative regulatory role for motif 1.

3) A general concern is that when using a transient overexpression system in the zebrafish larvae the potential variability of the expression of the individual construct and between individual cells may affects the conclusion on mRNA localization. The authors should comment on this.

4) For the evaluation of the mRNA localization in figure 4B there appear to be a correlation between high mRNA localization to the myelin membrane and low membrane expression of mScarlet-Caax. This could potential mean that the fact that the CAAX mediated localization requires initial association with endo-membranes prior to membrane targeting prevent proper membrane targeting for proteins encoded by efficient localized mRNAs especially if the mRNA expression is low. Can the authors exclude that this does not have an effect on the conclusion drawn on the efficiency of the mRNA targeting.

5) With respect to the data presented in Figure 8 for other researches to gain an insight in the specific transcript encoding motif 2 the raw data from the analysis should be include as supplementary material.

6) In the material and methods section on Page 19 line 463-464 the authors state that they have not been able to verify the sequence of the cloned dlg1 3'UTR and do therefor not known whether the sequencing between 54 and 775 of the 3'UTR match the database sequence. To be able to make any conclusion based on the experiments performed with this construct verification of the sequenced used is required. If this is not possible, the data on this construct should be removed to avoid future confusion, as it is not critical for the conclusions of the paper.

Rev. 3:

This manuscript by the Appel group presents the "Identification of 3'UTR motifs important for the distal localisation of myelin mRNAs to local myelin sheaths". Using the MS2 system for live imaging of mRNA localisation in zebrafish and single molecule fluorescent in situ hybridisation, the authors first confirm the validity of their assays by showing that myelin basic protein mRNA is detectable in myelin sheaths and highly enriched when compared to a control mRNA (Figures 1 and 2). Using co-labelling with actin-reporter constructs, the authors further reveal that mbp mRNAs predominantly localise to the leading edge of the myelin sheath (the growth zone of the myelin sheath with regard to its length and thickness, Figure 3). The authors then carried out a bioinformatic analysis to identify additional candidate mRNAs that may localised to myelin, but which are not classical myelin proteins (Figure 4). They present six of these candidates and show that three of them (eif4eb2, fmr1, irrtm1) accumulate in distal myelin processes using the MS2 system for live RNA imaging (Figure 4) and single molecule FISH (Figure 5). Further analysis of the respective 3' UTR lead to the identification of three different consensus motifs, and the selective deletion of these motifs reduced distal mRNA accumulation in their MS2 mRNA localisation reporters (Figure 6). In a next step, the candidate consensus motifs were attached to a control mRNA, which showed that the 3' UTR motifs of mbp mRNA are sufficient for distal mRNA targeting, but not the 3' UTR motifs of the other candidate mRNAs identified here (Figure 7). Lastly, the authors provide additional bioinformatic analysis of myelin mRNAs derived from existing datasets to show that one of the 3' UTR consensus motifs (motif 2) was enriched in the 3'UTRs of about half of all mRNAs found in myelin.

Overall a very nice manuscript with high-quality images, and a range of different methods approaches and newly generated reagents. The manuscript is also well written and easy to follow (apart from lacking explanations of some abbreviations which may not be obvious to most readers). Targeted transport and local translation of mRNAs are important regulators of cell function, particularly in the context of local control of cell function during adaptive processes. The work presented here will make a significant contribution to the mechanisms underlying the control of distal mRNA targeting for the regulation of local myelin growth if the authors can address the following points:

Main points:

- The manuscript is set up with the overall idea, that specific targeting of mRNA to individual sheaths may contribute to local control of myelin growth, and therefore be one of the reasons why some myelin sheaths are shorter and others maybe longer. It would be good if the authors could also provide experimental evidence for this theory, because the provision of material for new myelin may also depend on other mechanisms such as translation efficiency and/or mRNA stability, as the authors also discuss on p15 (lines 362-364). If it is not be possible to directly test this experimentally here, I wonder if the authors could provide correlative evidence between sheath length and mRNA abundance? Are myelin sheaths overall longer when they contain a higher density of local mRNAs? Would it be possible to retrieve quantitative information from the existing data that the authors have at hand? For example, the cell from the experiments in Figure 3D has sheaths with and without mRNA accumulations.

- The consensus motif 2 identified in the different 3' UTRs seems to be the most abundant one that is present in myelin mRNAs, as well as in additional mRNAs of non-classical myelin components, but which also localise to myelin. In Figure 6 the authors have systematically deleted each of these consensus motifs, which showed that all of them are necessary, at least to some extent. However, Figure 7 shows that only the motifs 2/3 of mbp mRNA were sufficient to induce distal localisation of a control mRNA, whereas the motifs 1/2/3 of eif4eb2 and fmr1 were not. I find these data difficult to interpret as the manuscript stands now. To my understanding there is either something special about the motifs 2/3 (or motif 2 or 3) in 3'UTR of mbp, which makes them different from the ones of eif4eb2 and fmr1. This could be tested by for example replacing the motifs 2 and/or 3 of eif4eb2 or fmr1 with the ones of the mbp 3' UTR. Alternatively, additional unidentified sequences may be crucial too. Independent of the exact outcome, additional experimental information would help clarify the significance of the individual motifs.

- It is difficult to see the differences in distal mRNA localisation for some cells in the images shown in Figure 4. Fmr1 is obvious, but, judging from the images, why would cadm1b be evenly distributed in the cell while eif4ebp2 has greater intensities in the sheaths? I understand that the quantification provides according significant differences, but the images don't make this appear obvious. Here, the smFISH data presented in Figure 5 are much more convincing than MS2 data in Figure 4. However, the smFISH are entirely qualitative at present. If the authors could provide according quantifications, including quantifications of an example mRNA which does not specifically localise to sheaths like cadm1b, this would greatly strengthen the validity of the MS2 approach, and well as the authors conclusions.

Minor points:

- Colour coding in Figures 1C-F, 4B, 6C, 7B. For faint fluorescence, blue over black is very hard to discriminate, at least for my eyes. This weakens the visual effects of the images. And for bright fluorescence, yellow/white over grayscale is very difficult to discriminate too, so that the merged channels don't provide much additional information in their current form.

- Many of the abbreviations used are not written out. This makes it difficult to immediately grasp the experimental approach as it is described in the results section.

- I would suggest to swap panels of Figure 3E and 3F, because 3F is mentioned before E in the main text.

Rev. 4:

Localised mRNAS and local translation into protein is increasingly recognized as a guiding principle in many biological systems. In particular neurons utilize this to localize and locally translate many mRNAS for example, at synapses. It is increasingly recognized that this principle is used by oligodendrocytes to sort mRNAs into forming myelin and indeed mbp mRNA was discovered early on to be a pertinent example here. The present paper is investigating the identity of 3´UTR motifs required to localize the abundant number of mRNAs found in myelin. They utilise the power of living zebra fish combined with the MS2 system developed by the lab of Robert Singer for tagging mRNAs, quantifying and visualizing their subcellular localisation after attachment to a fluorescent EGFP reporter binding the MS2 loops. Their EGFP reporter is driven by a sox10 promoter to allow expression in oligodendrocytes and has an NLS sequence added to reduce non-specific fluorescence.

The authors initially tested their tools using the well-characterised 3´UTR of mbp mRNA coupled to a Scarlet-Caax reporter which is translated into protein. As a control they utilized the 3´UTR of SV40. This also allowed them to define the optimal time window (4dpf) for visualising endogenous mbp mRNA using single molecule FISH. They observed that this co-localised with a F-actin reporter indicating that mbp mRNA occupies the leading edge of the forming myelin sheath. In contrast, the control sv40 3´UTR did not enrich at the leading edge.

They subsequently investigated the published data of mRNAs localised in murine myelin from P18 brains. The myelin was obtained using a classical biochemical purification of myelin over density gradients, which does not exclude a degree of contamination, for example, with cell processes. They selected candidate 3´UTRs using the GO terms oligodendrocyte, myelin, translation and synapse, also looking for zebra fish orthologs. They came up with six candidate genes: cadm1b, cyfip1, dlg1, eif4ebp2, fmr1 and lrrtm1.

Cadm1 and Irrtm1 are both transmembrane proteins which would be synthesised on rough ER. In fact, their latest very interesting Nat. Comm. paper has examined Cadm1 in detail. On the whole, it is mRNAs which are translated on free ribosomes that exhibit long 3´UTRs directing localization and local translation in cellular compartments. It is thus not surprising that the 3´UTRs of Cadm1 mRNA does not promote localisation to myelin. I would recommend that this point should be brought out and discussed more clearly in the Discussion. In fact, I would have selected initially other candidates that coded for proteins translated on free ribosomes instead of these two genes, although they clearly serve as good negative controls. Interestingly though, Irrtm1 3´UTR did localise to myelin sheaths! Perhaps they could discuss this point further.

They succeeded in identifying three new motifs in the 3´UTRs of three (mbp, eif4ebp2 and fmr1) of the localised mRNAs which contain cis-regulatory motifs. These are absent fromn the non-localised mRNAs and also not present in Irrtm1 3`UTR. One of these motifs is found abundantly in the myelin transcriptome.

This is thus an important paper, using state of the art technology as it demonstrates in vivo sorting principles of mRNAs to myelin and has identified new motifs playing a role here.

---

## [Decision Letter · Decision Letter 2]

7 Aug 2020

Dear Dr Appel,

I am writing on behalf of my colleague Ines Alvarez-Garcia, who is on annual leave.

Thank you very much for submitting a revised version of your manuscript "Identification of 3’ UTR motifs required for mRNA localization to myelin sheaths in vivo" for consideration as a Research Article at PLOS Biology. This revised version of your manuscript has been evaluated by the PLOS Biology editors and by the original Academic Editor and reviewers 2 and 3.

In light of the reviews (below), we will not be able to accept the current version of the manuscript. Reviewer 3 still thinks that some of your conclusions are not yet fully supported by experimental data. Having discussed these comments with the Academic Editor, and considering the support from the other reviewers, we are willing to offer you one last opportunity to address the lingering concerns. Please note that we will not be able to make any decision about publication until we have seen the revised manuscript and your response to the reviewers' comments. Your revised manuscript is also likely to be sent for further evaluation by the reviewers.

We expect to receive your revised manuscript within 3 months. 

Please email us (plosbiology@plos.org) if you have any questions or concerns or would like to request an extension. At this stage, your manuscript remains formally under active consideration at our journal; please notify us by email if you do not intend to submit a revision so that we may end consideration of the manuscript at PLOS Biology.

**IMPORTANT - SUBMITTING YOUR REVISION**

Your revisions should address the specific points made by reviewer 3. Please submit the following files along with your revised manuscript:

*Re-submission Checklist*

*Published Peer Review*

*PLOS Data Policy*

*Blot and Gel Data Policy*

Sincerely,

Gabriel Gasque, PhD,

Senior Editor

on behalf of 

Ines Alvarez-Garcia, PhD,

Senior Editor,

ialvarez-garcia@plos.org,

PLOS Biology

REVIEWS:

Reviewer #2: The additional data analysis and changes made to the text of the manuscript address my concerns in a satisfactory manner, I therefore recommend the manuscript for publication in PLoS Biology. 

Reviewer #3: In my previous review, I had raised major and minor points relating to the conclusions that can be drawn from the data as they are presented. These concerns remain because the authors do not provide additional data to address my questions.

1. I asked for (at least correlative) evidence that differences in mRNA abundance correlate with the properties of the respective myelin sheath (e.g. length). This point has not been addressed in the revised version so that the reader is left with notion that distal mRNA transport is crucial for local regulation of myelin growth (see introduction), but no evidence is provided that this is actually the case in the system presented here.

2. I had asked for additional experimental information about the necessity and sufficiency about motifs 2 and 3, because although being present in many distally transported transcripts, only the motifs derived from the mbp 3'UTR seem sufficient for distal mRNA targeting. The authors now discuss possible explanations for this circumstance, which may all be true. However, the significance of the consensus motifs remains very difficult to assess.

---

## [Editor Report · Decision Letter 3]

26 Nov 2020

Dear Dr Appel,

Thank you for submitting your revised Research Article entitled "Identification of 3’ UTR motifs required for mRNA localization to myelin sheaths in vivo" for publication in PLOS Biology. I have now obtained advice from the Academic Editor and have discussed the revision with the team of editors. 

We're delighted to let you know that we're now editorially satisfied with your manuscript. However before we can formally accept your paper and consider it "in press", we also need to ensure that your article conforms to our guidelines. A member of our team will be in touch shortly with a set of requests. As we can't proceed until these requirements are met, your swift response will help prevent delays to publication. Please also make sure to address the data and other policy-related requests noted at the end of this email.

- a cover letter that should detail your responses to any editorial requests, if applicable

*Copyediting*

*Published Peer Review History*

*Early Version*

Sincerely,

Ines

--

Ines Alvarez-Garcia, PhD

Senior Editor,

PLOS Biology

Fig. 1E; Fig. 2C-E; Fig. 3E, F; Fig. 4D; Fig. 6D, E; Fig. 7D, E and Fig. 8E, F

---

## [Editor Report · Decision Letter 4]

22 Dec 2020

Dear Dr. Appel,

I am writing concerning your manuscript submitted to PLOS Biology, entitled “Identification of 3’ UTR motifs required for mRNA localization to myelin sheaths in vivo.”

We have now completed our final technical checks and have approved your submission for publication. You will shortly receive a letter of formal acceptance from the editor.

Kind regards,

PLOS Biology